# Whole blood transcriptional profiles and the pathogenesis of tuberculous meningitis

Hoang Thanh Hai[1†], Le Thanh Hoang Nhat[1†], Trinh Thi Bich Tram[1], Do Dinh Vinh[1], Artika P Nath[2], Joseph Donovan[1,3], Nguyen Thi Anh Thu[1], Dang Van Thanh[1], Nguyen Duc Bang[4], Dang Thi Minh Ha[4], Nguyen Hoan Phu[1,5], Ho Dang Trung Nghia[1,5,6], Le Hong Van[1], Michael Inouye[2,7], Guy E Thwaites[1,3], Nguyen Thuy Thuong Thuong[1,3]*

[1]Oxford University Clinical Research Unit, Ho Chi Minh City, Viet Nam; [2]Cambridge Baker Systems Genomics Initiative, Baker Heart and Diabetes Institute, Melbourne, Australia; [3]Centre for Tropical Medicine and Global Health, Nuffield Department of Medicine, University of Oxford, Oxford, United Kingdom; [4]Pham Ngoc Thanh Hospital, Ho Chi Minh City, Viet Nam; [5]Hospital for Tropical Diseases, Ho Chi Minh City, Viet Nam; [6]Pham Ngoc Thach University of Medicine, Ho Chi Minh City, Viet Nam; [7]Cambridge Baker Systems Genomics Initiative, Department of Public Health and Primary Care, University of Cambridge, Cambridge, United Kingdom

*For correspondence: thuongntt@oucru.org

[†]These authors contributed equally to this work

## eLife assessment

In this **valuable** study, the authors investigate the transcriptional landscape of tuberculous meningitis. They reveal potentially significant molecular differences contributed by HIV co-infection, and derive a prognostic model to predict mortality combining a gene expression signature with clinical parameters. Whilst some of the evidence presented is **compelling**, the bioinformatics analysis remains limited and cannot be used to make causal inferences and conclusions about immuno-pathogenesis for tuberculous meningitis. The work will be of broad interest to the infectious disease community however, further validation of the findings is critical for future utility.

**Abstract** Mortality and morbidity from tuberculous meningitis (TBM) are common, primarily due to inflammatory response to *Mycobacterium tuberculosis* infection, yet the underlying mechanisms remain poorly understood. We aimed to uncover genes and pathways associated with TBM pathogenesis and mortality, and determine the best predictors of death, utilizing whole-blood RNA sequencing from 281 Vietnamese adults with TBM, 295 pulmonary tuberculosis (PTB), and 30 healthy controls. Through weighted gene co-expression network analysis, we identified hub genes and pathways linked to TBM severity and mortality, with a consensus analysis revealing distinct patterns between HIV-positive and HIV-negative individuals. We employed multivariate elastic-net Cox regression to select candidate predictors of death, then logistic regression and internal bootstrap validation to choose best predictors. Increased neutrophil activation and decreased T and B cell activation pathways were associated with TBM mortality. Among HIV-positive individuals, mortality associated with increased angiogenesis, while HIV-negative individuals exhibited elevated TNF signaling and impaired extracellular matrix organization. Four hub genes—*MCEMP1, NELL2, ZNF354C,* and *CD4*—were strong TBM mortality predictors. These findings indicate that TBM induces a systemic inflammatory response similar to PTB, highlighting critical genes and pathways

related to death, offering insights for potential therapeutic targets alongside a novel four-gene biomarker for predicting outcomes.

## Introduction

Of 7.1 million new tuberculosis (TB) cases in 2019, tuberculous meningitis (TBM) is estimated to have developed in 164,000 adults, around 25% of whom were living with HIV (*Dodd et al., 2021*). TBM is the most severe form of TB, causing death or neurological disability in half of all cases. Overall, TBM mortality is around 25%, but rises to around 50% in those with HIV, with most deaths occurring in the first 3 months of treatment (*Dodd et al., 2021*; *Stadelman et al., 2020*).

The poor outcomes from TBM are strongly associated with the inflammatory response (*Wilkinson et al., 2017*; *Huynh et al., 2022*), with both a paucity and an excess of inflammation linked to death from TBM (*Marais et al., 2017*; *Rohlwink et al., 2017*; *Thuong et al., 2017*; *Thuong et al., 2019*; *van Laarhoven et al., 2017*; *van Laarhoven et al., 2019*; *Cresswell et al., 2019*). However, the mechanisms behind these observations remain uncertain. Immune responses in TBM are thought to be compartmentalized within the central nervous system. Studies have shown that immune cell counts, cytokine concentrations, metabolites and transcriptional responses differ between the peripheral blood and the cerebrospinal fluid (CSF) (*van Laarhoven et al., 2019*; *Rohlwink et al., 2019*; *Ardiansyah et al., 2023*). In adults with TBM, leukocyte activation is higher in the CSF than in peripheral blood, although a marked myeloid response in peripheral blood has been reported (*van Laarhoven et al., 2019*). Blood transcriptomic analysis has found increased neutrophil-associated transcripts and inflammasome signaling in those with HIV-associated TBM and immune reconstitution inflammatory syndrome (*Marais et al., 2017*). In children with TBM, whole blood transcriptional profiles showed increased inflammasome activation and decreased T-cell activation (*Rohlwink et al., 2019*). Taken together, these studies suggest the inflammatory response associated with TBM may have a greater systemic component than originally thought and its characterization may help define the immune mechanisms leading to death and disability.

The accurate and early detection of patients at highest risk of complications and death from TBM may help to target treatment for those most in need. The British Medical Research Council (MRC) grades have been used to categorize TBM severity for almost 80 years (*Streptomycin in Tuberculosis Trials Commitee, Medical Research Council, 1948*), and the system strongly predicts TBM mortality (*Thao et al., 2018*). Previously, we and others developed new prognostic models from studies of 1699 adults with TBM using clinical and laboratory parameters, including MRC grade (*Huynh et al., 2022*; *Thao et al., 2018*). These models predicted outcomes more accurately than MRC grade alone. However, they might be improved by measures of host inflammatory response. Host-based peripheral blood gene expression analysis has been used to identify active or progressive pulmonary TB and in pulmonary TB treatment monitoring (*Sweeney et al., 2016*; *Penn-Nicholson et al., 2020*), but has yet to be applied to TBM.

In the current study, we investigated whole blood RNA sequencing (RNA-seq) transcriptional profiles in 281 Vietnamese adults with TBM, 295 with pulmonary TB (PTB), and 30 healthy controls. Our objective was to use weighted gene co-expression network analysis, an unbiased and well-evaluated approach, to identify the biological pathways and hub genes associated with TBM pathogenesis and assess the predictive value of gene expression for early mortality from TBM.

## Results

### Characteristics and outcomes of the cohorts

Four RNA-seq cohorts (all ≥18 years) were used in the study, representing a total of 606 participants. The characteristics of these cohorts are provided in *Table 1*. There were 281 adults with TBM; 207 HIV-negative and 74 HIV-positive. In the HIV-negative TBM adults, the median age was 46 years (IQR 34, 58), 127 (61%) were male, and the median Body Mass Index (BMI) was 20.0 (IQR 18.2, 22.3). HIV-positive TBM were more likely than HIV-negative TBM to be male, younger, have lower BMI, have previously received TB treatment, and to have microbiologically confirmed TBM. Total white cell counts in blood and CSF in HIV-positive TBM were lower than in HIV-negative TBM. Median CD4 cell counts in HIV-positive TBM was 67 cells/mm$^3$ (IQR 19, 124) and 28 (39%) were under antiretroviral

**eLife digest** Tuberculous meningitis is a dangerous condition caused by the bacteria responsible for tuberculosis spreading from the lungs to the brain. It affects more than 150,000 adults a year worldwide, and results in death or brain damage in half of all patients. People living with HIV are particularly at risk for negative outcomes.

These severe forms of tuberculous meningitis may be linked to the immune system becoming overactive while trying to fight the disease and harming the brain in the process. Detecting this hyperinflammation via blood sample analyzes has remained challenging so far, as traditional approaches can only offer partial information on the inflammatory response.

In response, Hai, Nhat et al. took advantage of new genetic approaches to examine the expression of around 20,000 genes in the blood of HIV-positive and HIV-negative patients with tuberculous meningitis or lung tuberculosis, as well as in healthy individuals. Identifying which genes are more or less expressed in the different groups of volunteers can help to better understand the mechanisms associated with tuberculous meningitis, particularly in its most dangerous forms. Such analysis could also allow scientists to pinpoint which genes to monitor to efficiently detect patients at higher risk of severe complications.

The results show that tuberculous meningitis mortality was associated with a distinct pattern of immune cell response; white blood cells known as neutrophils were increasingly activated while T and B cells showed decreased activity. Increased mortality was also linked to different patterns of gene activity between patients living with or without HIV. Overall, inflammatory genes were more activated in HIV-positive tuberculous meningitis patients than in their HIV-negative counterparts. Finally, Hai, Nhat et al. found that the blood activity levels of just four specific genes formed a signature associated with increased risk of death from tuberculous meningitis.

In the future, medical professionals may be able to use this signature to rapidly identify patients who require intensive care and more specialized treatments. The findings also reveal immune system processes and molecules that may serve as potential drug targets for future therapies against this disease.

therapy. The PTB cohort consisted of 295 HIV-negative adults with the median age of 44 years (IQR 31, 52), 228 (77%) were male, the median of BMI was 19.4 (IQR 17.7, 21.6) and 129 (48%) had pulmonary cavities on chest X-ray. Of the 30 healthy controls, 11 (37%) were male, and the median age was 33 (IQR 29, 37). In real-time quantitative polymerase chain reaction (qPCR) validation cohort, 132 HIV-negative TBM adults have similar characteristics as HIV-negative TBM RNA-seq cohort (*Table 1*).

The clinical variables associated with three-month mortality of the 281 adults with TBM in RNA-seq cohort are given in *Table 2*. The discovery (n=142) and validation (n=139) cohorts had similar characteristics (*Table 2*). 47.3% (133/281) had definite TBM (*Marais et al., 2010*), with microbiologically confirmed disease, accounting for 45.9% (101/220) of survivors and 52.4% (32/61) of those who died. The overall three-month mortality rate was 21.7% (61/281) for TBM regardless of HIV status: 16.4% (34/207) in HIV-negative and 36.5% (27/74) in HIV-positive (p<0.001). We did not observe differences in mortality by sex, age and diagnostic category. Greater disease severity, MRC grades 2 and 3 at enrolment, was associated with increased mortality compared to grade 1 (p<0.001). In those who died, CSF and peripheral blood neutrophil counts were higher and peripheral blood lymphocyte count lower, compared to those who survived.

In qPCR validation cohort (n=132), 3-month mortality rate was 28.8% (38/132) and those who died associated with older age, greater disease severity, lower number of CSF leukocytes, lymphocytes and neutrophils, but did not differ in number of peripheral blood neutrophils (*Supplementary file 1A*).

## Whole blood transcriptional profiles of the four RNA-seq cohorts

We analyzed the whole blood transcriptomics, using bulk RNA sequencing from 606 participants in the 4 cohorts. On average 35.1 million reads/sample was obtained with 89.4% reads mapping accuracy to human reference genome (GRCh.38 release 99) and 65.4% reads were uniquely mapped. The study objectives and cohorts flow are presented in *Figure 1*. Principal component analysis on the transcriptomic data of 20,000 genes across 4 studies showed different profiles between healthy controls

**Table 1.** Baseline characteristics of TBM, PTB, and healthy controls.

| | RNA-seq cohorts | | | | | | | | | qPCR validation cohort | |
| | HIV-negative TBM n = 207 | | HIV-positive TBM n = 74 | | PTB n=295 | | Healthy controls n = 30 | | HIV-negative TBM n = 132 | |
| Characteristics | n | Summary | n | Summary | n | Summary | n | Summary | n | Summary |
|---|---|---|---|---|---|---|---|---|---|---|
| Age (years) | 207 | 46 (34, 58) | 74 | 34 (29, 40) | 295 | 44 (31, 52) | 30 | 33 (29, 37) | 132 | 48 (35, 60) |
| Male sex | 207 | 127 (61) | 74 | 56 (76) | 295 | 228 (77) | 30 | 11 (37) | 132 | 84 (65) |
| BMI (kg/m2) | 205 | 20.0 (18.2, 22.3) | 72 | 19.3 (17.2, 20.4) | 295 | 19.4 (17.7, 21.6) | | | 132 | 20.0 (18.2, 22.3) |
| Symptom duration (days) | 207 | 14 (11, 20) | 73 | 16 (10, 30) | 294 | 20 (10, 30) | | | 132 | 16 (13, 24) |
| History of TB treatment | 204 | 5 (2.5) | 74 | 15 (20) | 295 | 100 (34) | | | 130 | 12 (9.2) |
| Glasgow coma score | 207 | 14 (12, 15) | 72 | 14 (13, 15) | | | | | 132 | 14 (13, 15) |
| Cavity chest X-ray | | | | | 270 | 129 (48) | | | | |
| TB microbiological tests | | | | | | | | | | |
| MGIT culture positive | 199 | 50 (25) | 69 | 41 (59) | 295 | 279 (95) | | | | |
| Xpert/Ultra positive | 198 | 42 (21) | 70 | 39 (56) | 295 | 287 (97) | | | | |
| Microscopy positive | 205 | 48 (23) | 67 | 35 (52) | 203 | 169 (83) | | | 105 | 18 (17) |
| Blood (10^6 cells/ml) | | | | | | | | | | |
| Leucocyte count | 204 | 9.4 (7.0, 11.9) | 74 | 6.4 (5.0, 9.2) | 242 | 9.2 (7.4, 11.4) | 26 | 6.4 (5.6, 7.2) | 129 | 10.0 (7.7, 12.4) |
| Neutrophil count | 204 | 7.1 (4.8, 9.1) | 74 | 5.0 (3.3, 6.9) | 241 | 6.1 (4.7, 8.2) | 26 | 3.4 (3.1, 4.1) | 129 | 7.8 (6.9, 8.5) |
| Lymphocytes count | 204 | 1.2 (0.9, 2.0) | 74 | 0.7 (0.4, 1.2) | 242 | 1.9 (1.4, 2.3) | 26 | 2.2 (1.9, 2.6) | 129 | 1.2 (0.7, 1.8) |
| CSF (10^3 cells/ml) | | | | | | | | | | |
| Leucocyte count | 207 | 142 (19, 323) | 73 | 124 (10, 453) | | | | | 106 | 122 (38, 328) |
| Neutrophil count | 207 | 0 (0, 39) | 73 | 17 (0, 144) | | | | | 60 | 20 (3, 73) |
| Lymphocyte count | 207 | 106 (18, 223) | 73 | 58 (10, 216) | | | | | 106 | 95 (79, 100) |
| CD4 cell count (cells/mm3) | | | 71 | 67 (19, 124) | | | | | | |
| Antiretroviral therapy | | | 70 | 28 (39%) | | | | | | |
| HIV load (10^3 cells/ml) | | | 73 | 77.3 (0.8, 672) | | | | | | |

Values were displayed as median (1st and 3rd interquartile) for continuous variables and frequency (%) for categorical variables.

TBM = PTB = tuberculosis, andCSF = TBM = PTB = tuberculosis, and CSF = TBM = tuberculous meningitis, PTB = pulmonary tuberculosis, and CSF = cerebrospinal fluid.

from both PTB and TBM cases (*Figure 2A*). The PTB profile substantively overlapped with TBM, with some separation between HIV-negative and HIV-positive TBM.

Enrichment scores from single sample gene set enrichment analysis (ssGSEA), which based on expression rank of genes relevant to pathways, were measured in some pathways already known to be important mediators of TB or TBM pathogenesis (*Figure 2B–G*). As anticipated, inflammatory response, cytokine signaling, interferon signaling, TNF signaling, and inflammasome activation pathways, were enriched in PTB and TBM cohorts as compared to healthy controls. In TBM, enrichment of genes in these pathways were generally higher in HIV-positive than in HIV-negative individuals.

## Transcriptional gene modules associated with TBM severity and mortality

Transcriptional profiles associated with TBM mortality were generated by identifying differentially expressed genes. Of the top 20,000 genes with most variation, we observed 724 (3.6%) genes that were differentially expressed (FDR <0.05, absolute fold change (FC) >1.5) in all those with TBM (*Figure 3A*). Next, we applied weighted gene co-expression network analysis (WGCNA) to 5000 most variable genes from 281 TBM samples (n=207, HIV-negative; n=74, HIV-positive) to define clusters of highly correlated genes (modules) associated with TBM severity and mortality. Gene modules are

**Table 2.** Association between baseline clinical characteristics with TBM mortality in RNA-seq cohorts.

| Characteristics | All TBM | | | | | | Discovery cohort | | Validation cohort | |
| --- | --- | --- | --- | --- | --- | --- | --- | --- | --- | --- |
| | n | Survival n=220 | Death n=61 | HR* | 95% CI | p value* | Survival n=111 | Death n=31 | Survival n=109 | Death n=30 |
| Male sex | 281 | 144 (65) | 39 (64) | 0.92 | 0.55, 1.56 | 0.3 | 70 (63) | 19 (61) | 74 (68) | 20 (67) |
| Age (years) | 281 | 41 (32, 53) | 39 (30, 60) | 1.01 | 0.99, 1.02 | 0.8 | 41 (32, 53) | 47 (33, 64) | 41 (32, 53) | 35 (29, 59) |
| HIV infection | 281 | 47 (21) | 27 (44) | 2.34 | 1.41, 3.89 | <0.001 | 24 (22) | 14 (45) | 23 (21) | 13 (43) |
| †Diagnostic category | 280 | | | | | | | | | |
| definite TBM | | 101 (46) | 32 (52) | | | | 55 (50) | 17 (55) | 46 (42) | 15 (50) |
| possible TBM | | 43 (20) | 6 (9.8) | 1.01 | 0.59, 1.72 | 0.9 | 20 (18) | 2 (6.5) | 23 (21) | 4 (13) |
| probable TBM | | 75 (34) | 23 (38) | 0.50 | 0.21, 1.19 | 0.12 | 35 (32) | 12 (39) | 40 (37) | 11 (37) |
| ‡MRC grade | 281 | | | | | | | | | |
| grade 1 | | 114 (52) | 7 (11) | | | | 58 (52) | 3 (9.7) | 56 (51) | 4 (13) |
| grade 2 | | 100 (45) | 35 (57) | 4.96 | 2.20, 11.2 | <0.001 | 51 (46) | 18 (58) | 49 (45) | 17 (57) |
| grade 3 | | 6 (2.7) | 19 (31) | 26.4 | 11.0, 63.2 | <0.001 | 2 (1.8) | 10 (32) | 4 (3.7) | 9 (30) |
| Blood ($10^6$ cells/ml) | | | | | | | | | | |
| Leukocyte count | 278 | 8.1 (6.2, 10.7) | 10.1 (6.7, 12.2) | 1.57 | 1.04, 2.38 | 0.033 | 7.8 (6.2, 10.6) | 8.6 (6.4, 12.0) | 8.4 (6.4, 10.7) | 10.8 (7.6, 12.4) |
| Neutrophil count | 278 | 6.0 (4.0, 8.1) | 8.0 (5.0, 10.3) | 1.83 | 1.29, 2.59 | <0.001 | 5.9 (4.0, 8.1) | 6.7 (4.9, 9.6) | 6.4 (4.0, 8.2) | 9.0 (5.8, 10.3) |
| Lymphocyte count | 278 | 1.2 (0.8, 1.9) | 0.9 (0.5, 1.2) | 0.70 | 0.56, 0.88 | 0.002 | 1.2 (0.7, 2.0) | 1.0 (0.6, 1.3) | 1.3 (0.9, 1.9) | 0.8 (0.5, 1.2) |
| CSF ($10^3$ cells/ml) | | | | | | | | | | |
| Leukocyte count | 280 | 129 (19, 340) | 148 (15, 360) | 0.99 | 0.91, 1.09 | 0.9 | 130 (17, 348) | 148 (15, 444) | 124 (27, 335) | 146 (25, 283) |
| Neutrophil count | 117 | 78 (28, 192) | 135 (54, 402) | 1.23 | 1.01, 1.49 | 0.040 | 64 (30, 228) | 260 (163, 516) | 90 (29, 161) | 75 (50, 173) |
| Lymphocyte count | 280 | 101 (18, 229) | 91 (14, 176) | 0.95 | 0.86, 1.05 | 0.3 | 104 (16, 208) | 123 (10, 214) | 97 (21, 248) | 86 (14, 147) |

Values were displayed as median (1st and 3rd interquartile) for continuous variables and frequency (%) for categorical variables.

TBM = CSF = TBM = tuberculous meningitis, CSF = cerebrospinal fluid.

*Association of the corresponding variables with three-month mortality using a Cox regression model. Hazard ratio (HR) and p-value from the Cox regression model were presented in the table. For blood and CSF cells, the HR was calculated for each increase in $\log_2$ units.

†Diagnostic categories were assigned according to the consensus case definition (**Marais et al., 2010**). Definite TBM refers to cases that are microbiologically confirmed TB by microscopy, culture and Xpert using CSF samples.

‡MRC grade denotes modified British Medical Research Council criteria (**Streptomycin in Tuberculosis Trials Commitee, Medical Research Council, 1948**).

clusters of genes that have highly interconnected expression, usually related to their biological functions. Hub genes are genes with high connectivity to other genes within a respective module. First, we used WGCNA to construct a network of gene modules in the discovery cohort. Then we validated the presence of these transcriptional modules in the validation cohort, labelling the modules with different colors. In the discovery cohort (n=142), 15 modules were identified overall, consisting of 44–1350 genes per module (*Figure 4—figure supplement 1*). All 15 modules were preserved in the validation cohort (n=139) through the preservation analysis (*Figure 4—figure supplement 2*).

The associations between the 15 modules and TBM severity and mortality are presented in *Figure 4* for both the discovery and validation cohorts. Modules were linked to each other in a hierarchical structure, with major biological processes annotated. Associations of the modules with TBM disease severity (MRC grade) at baseline were estimated by Spearman correlations between MRC grade and the first principle component (PC1) of each module. Similarly, associations of the modules with mortality were measured by hazard ratio (HR) per increase 1/10 unit of PC1 using Cox regression model adjusted for age, HIV status and dexamethasone treatment (*Figure 4*) in both discovery and validation cohorts.

Of the 15 preserved modules, five modules were significantly associated with mortality in the discovery and validation cohorts, with false discovery rate (FDR) <0.05 (*Figure 4B*). These five modules were separated into two big module clusters. The first cluster contained the blue module (n=799 genes), involved in inflammatory and innate immune responses, and the cyan module (n=44 genes) with unknown biological function. These modules were upregulated in those who died, as

Overview of blood transcriptional profiling and
Comparisons of known immune pathways & genes associated with TB pathogenesis
in 4 RNA-seq cohorts: 207 HIV-negative TBM, 74 HIV-positive TBM, 295 PTB and 30 HC

**Figure 2**

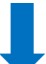

Defining of blood transcriptomic signatures associated with TBM mortality by
gene modules, hub genes and pathways  in all TBM and TBM stratified by
HIV status (n=281) and
Validation of hub genes in qPCR validation cohort (132 HIV-negative TBM)

**Figures 3, 4, ,5, 8; Table 3**

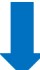

Comparisons of hub genes and pathways associated with TBM mortality
through gene expression level and pathway enrichment score
in 4 RNA-seq cohorts: HIV-negative TBM, HIV-positive TBM, PTB and HC

**Figures 6, 7, 9**

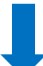

Prediction models for TBM prognostics
in 2 cohorts: TBM HIV-negative, TBM HIV-positive and
Evaluate prognostic signatures in qPCR validation cohort (132 HIV-negative TBM)

**Table 4**

**Figure 1.** Objectives and cohorts flow. TBM: TB meningitis, HIV: human immunodeficiency virus, PTB: pulmonary TB, HC; healthy controls.

The online version of this article includes the following figure supplement(s) for figure 1:

**Figure supplement 1.** Analysis workflow diagram.

**Figure supplement 2.** Batch correction for RNA-seq normalized count data.

shown in the heat-map in *Figure 4A* (HR: 3.0 and 2.2 for the blue module, and 2.1 and 1.7 for the cyan module, FDR <0.05 for all comparisons). The black (n=207 genes), brown (n=698 genes), and red (n=229 genes) modules were in the second cluster and were generally involved in adaptive immunity including T and B cell signaling pathways. These three modules were down-regulated in those who died (HR: 0.43 and 0.36 for brown, 0.46 and 0.39 for red, 0.59 and 0.49 for black; *Figure 4A*).

It is known that TBM MRC grade before treatment initiation strongly predicts outcome from TBM. Here, we investigated correlations between each module and MRC grade and their association with mortality. The pink module, involved in hemostasis and platelet activation, was positively correlated with MRC grade, but not mortality. Of the five modules associated with death from TBM, all were correlated with MRC grade (*Figure 4A*). Four of these five modules, were enriched for immune responses.

## Transcriptional hub genes associated with TBM severity and mortality

We next identified hub genes within the four biologically functional modules associated with TBM mortality in both the discovery and validation cohorts. Hub genes showed higher connectivity within

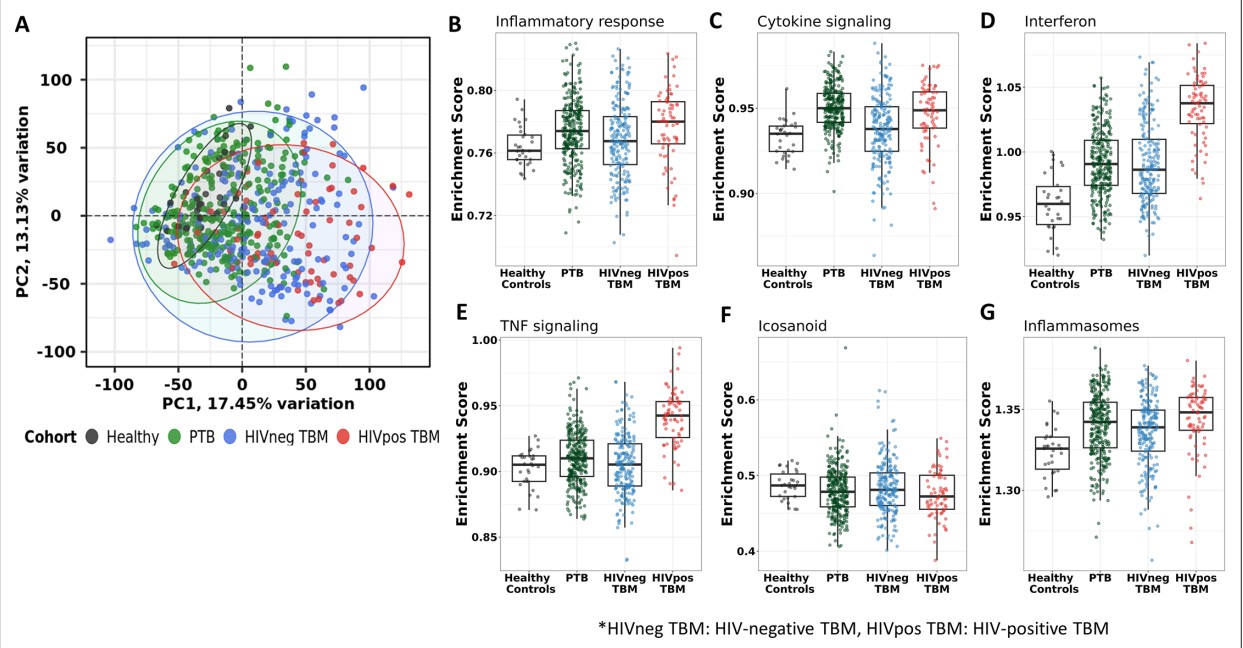

**Figure 2.** Blood transcriptomic profiles of four cohorts: healthy controls (n=30), PTB (n=295), HIV-negative TBM (n=207), and HIV-positive TBM (n=74). (**A**) Principle component analysis (PCA) of whole transcriptomic profile of HC, PTB and TBM with and without HIV. Each symbol represents one individual with color coding different cohorts. The x-axis represents principle component (PC) 1, while y-axis represents PC2. (**B–G**) Enrichment scores of known innate immunity pathways associated with TBM pathogenesis. Pathway enrichment scores were calculated using single sample GSEA algorithm (ssGSEA) (**Barbie et al., 2009**). Each dot represents one participant. The box presents median, 25th to 75th percentile and the whiskers present the minimum to the maximum points in the data.

the modules, and stronger association with TBM mortality as compared to less connected genes within a module (**Figure 5—figure supplement 1**). Seven hub genes (*ETS2, PGD, UPP1, CYSTM1, FCAR, KIF1B,* and *MCEMP1*) were upregulated in death, all from the acute inflammation (blue) module. Hub genes from the brown, red and black modules were downregulated in death and were involved in adaptive immune response. Ten hub genes associated with mortality (*CD96, TNFRSF25, TBC1D4, CD28, ABLIM1, RASGRP1, NELL2, TRAF5, TESPA1, TRABD2A*) were from the brown module, three (*EVL, PLCG1, NLRC3*) from the red, and six (*CD2, CD247, TGFBR3, ARL4C, KLRK1, MATK*) from the

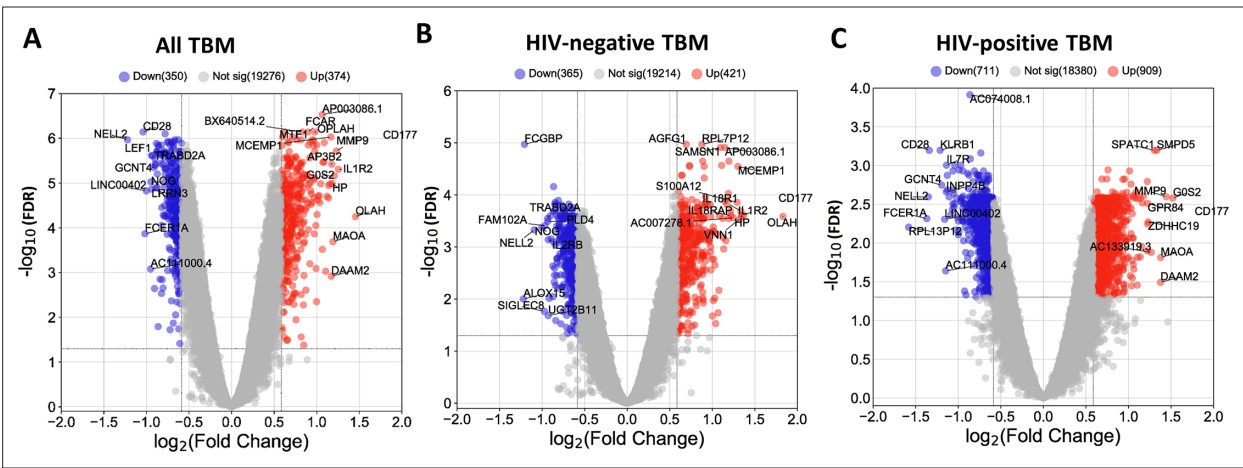

**Figure 3.** Blood transcriptomic profiles of three-month mortality at baseline in all TBM and TBM stratified by HIV status. Volcano plot showed differentially expressed (DE) genes by fold change (FC) between death and survival in all TBM (**A**), HIV-negative (**B**) and HIV-positive TBM (**C**). Each dot represents one gene. The x-axis represents log₂ FC. The y-axis showed –log₁₀ FDR of genes. DE genes were colored with red indicating up-regulated, blue indicating down-regulated genes which having fold discovery rate (FDR) <0.05 and absolute FC >1.5.

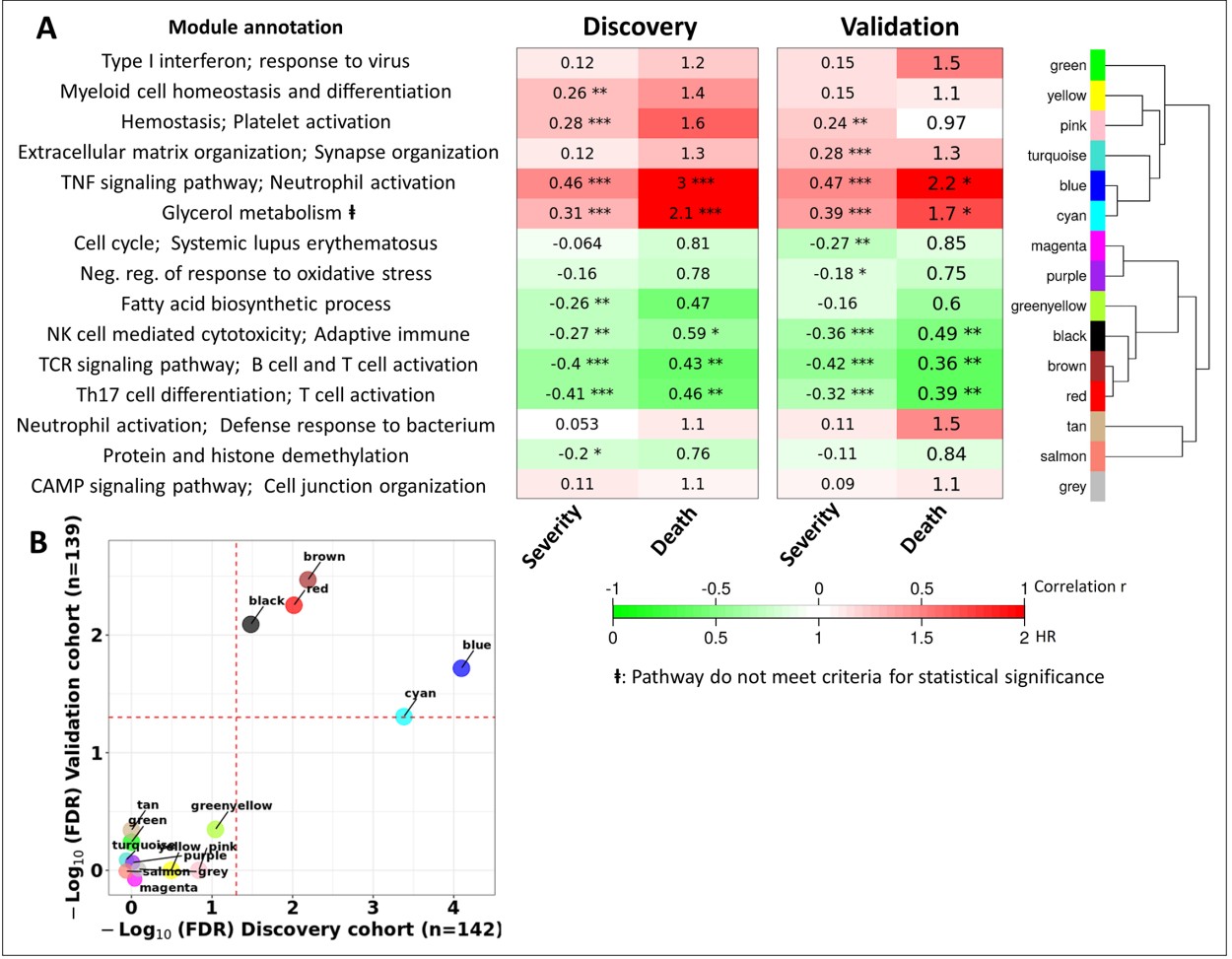

**Figure 4.** Blood transcriptional modules associated with mortality in TBM. (**A**) Associations between WGCNA modules with two clinical phenotypes TBM disease severity (MRC grade) and three-month mortality in discovery and validation cohorts, and their associated biological processes. The heatmap showed the association between principle component 1 (PC1) of each module and the phenotypes, particularly Spearman correlation r for MRC grade and hazard ratio per increase 1/10 unit of PC1 (HR) for mortality. The HRs were estimated using a Cox regression model adjusted for age, HIV status and dexamethasone treatment. False discovery rate (FDR) corrected based on Benjamini & Yekutieli procedure, with significant level denoted as *<0.05, **<0.01 and ***<0.001. Gradient colors were used to fill the cell with green indicating negative r or HR <1, red color indicating positive r or HR >1. The order of modules was based on hierarchical clustering using Pearson correlation distance for module eigengene. On the left, biological processes, corresponding to modules, were identified using Gene Ontology and KEGG database. (**B**) Validation of the association between WGCNA modules and mortality in discovery and validation cohorts. X-axis represents –log₁₀ FDR in discovery cohort and Y-axis represents –log₁₀ FDR in validation cohort. Red dash lines indicate FDR = 0.05 as the threshold for statistically significant in both cohorts. Five modules (blue, brown, red, black and cyan) with FDR <0.05 were validated.

The online version of this article includes the following figure supplement(s) for figure 4:

**Figure supplement 1.** Construction of WGCNA in discovery cohort.

**Figure supplement 2.** Preservation of discovery modules in validation cohort.

black module (*Figure 5E–H*). For qPCR validation, available samples from HIV-negative TBM patients (n=132) were used to evaluate 11 hub genes selected from the blue and brown modules. 8/11 hub genes were found to be significantly associated with mortality, as determined univariate Cox regression (*Table 3*). Three genes (*FCAR*, *PGD*, *ETS2*) also found to be associated but in reverse direction.

We, next, examined patterns of shared and distinct gene expression of some hub across the four cohorts (healthy controls, PTB, and TBM with and without HIV-infection) to reveal disease progression and pathogenesis of different TB forms (*Figure 6*). There were two upregulated genes from the acute inflammation module (*FCAR* and *MCEMP1*) and six downregulated genes (*NELL2*, *TRABD2A*, *PLCG1*, *NLRC3*, *CD247*, and *MATK*) from the other three adaptive immunity modules. The patterns of up and

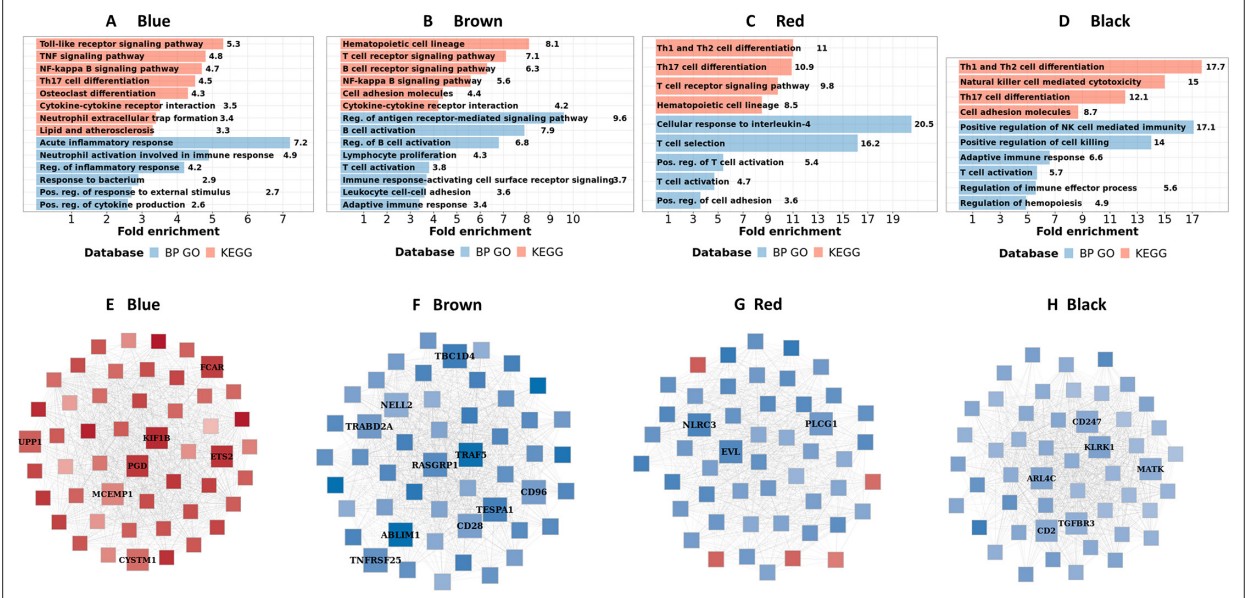

**Figure 5.** Biological processes, pathways and hub genes of validated modules associated with mortality. (**A–D**) showed biological processes and pathways identified in four mortality associated modules: blue, brown, red and black module, by over representation analysis (ORA). Bar plots show the top representative GO biological processes or KEGG pathways. The bars indicates biological processes or pathways having ORA FDR <0.05 and size corresponding to fold enrichment calculated as the ratio of gene number of pathway in the input list divided by the ratio of gene number of the pathway in reference. (**E–H**) showed gene co-expression networks and hub genes of blue, brown, red and black module, respectively. Each node represents one gene. Each edge represents the link between two genes. Hub genes were shown by bigger nodes and bold text. The gradient color of node corresponds to its HR per 1 log₂ unit increase in gene expression related to mortality, with red indicating HR >1, and blue HR <1.

The online version of this article includes the following figure supplement(s) for figure 5:

**Figure supplement 1.** Correlations between gene module membership and gene significance with mortality in four associated modules in discovery and validation cohorts.

**Table 3.** Validation of hub genes in PCR validation cohort.

| No. | Genes | RNA-seq cohort HIV-negative TBM (n=207) | | | qPCR validation cohort HIV-negative TBM (n=132) | | |
|---|---|---|---|---|---|---|---|
| | | HR | 95% CI | p value | HR | 95% CI | p value |
| 1 | MCEMP1 | 1.93 | 1.57, 2.37 | 4.07E-10 | 1.74 | 1.27, 2.37 | 4.99E-04 |
| 2 | FCAR | 2.37 | 1.70, 3.30 | 3.52E-07 | 0.59 | 0.40, 0.87 | 7.87E-03 |
| 3 | ETS2 | 2.65 | 1.85, 3.81 | 1.24E-07 | 0.27 | 0.16, 0.44 | 3.27E-07 |
| 4 | PGD | 2.65 | 1.84, 3.81 | 1.47E-07 | 0.18 | 0.09, 0.34 | 2.69E-07 |
| 5 | NELL2 | 0.56 | 0.45, 0.71 | 6.08E-07 | 0.80 | 0.66, 0.96 | 1.53E-02 |
| 6 | TRABD2A | 0.43 | 0.31, 0.58 | 4.94E-08 | 0.64 | 0.51, 0.8 | 1.05E-04 |
| 7 | TRAF5 | 0.31 | 0.2, 0.49 | 4.52E-07 | 0.47 | 0.36, 0.61 | 1.66E-08 |
| 8 | CD28 | 0.57 | 0.42, 0.76 | 1.42E-04 | 0.60 | 0.47, 0.76 | 2.83E-05 |
| 9 | TESPA1 | 0.36 | 0.25, 0.52 | 2.74E-08 | 0.65 | 0.48, 0.89 | 7.35E-03 |
| 10 | ABLIM1 | 0.31 | 0.2, 0.49 | 2.21E-07 | 0.47 | 0.36, 0.63 | 1.55E-07 |
| 11 | RASGRP1 | 0.45 | 0.31, 0.67 | 6.10E-05 | 0.59 | 0.45, 0.77 | 9.77E-05 |

Association of the hub genes with three-month mortality using a univariate Cox regression model. Hazard ratio (HR), 95% CI of HR and p-value from the Cox regression model were presented in the table. HR or 95% mean per increase 1 unit of log2 normalized expression of gene in RNA-seq cohort or decrease 1 unit of cycle threshold in qPCR validation cohort.

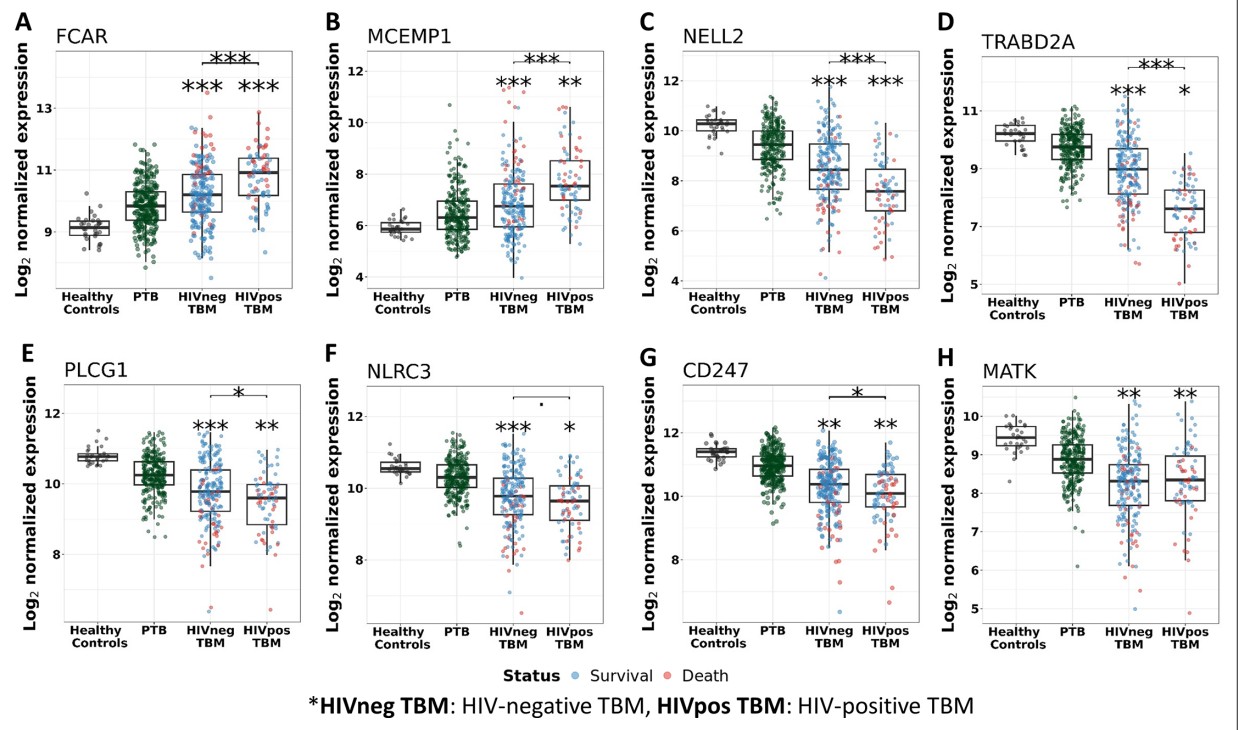

**Figure 6.** Gene expression of representative hub genes in healthy controls (n=30), PTB (n=295), HIV-negative TBM (n=207), and HIV-positive TBM (n=74). Each dot represents gene expression from one participant. (**A, B**) expression of *FCAR* and *MCEMP1* hub genes from the blue module. (**C, D**) expression of *NELL2* and *TRABD2A* hub genes from the brown modules. (**E, F**) expression of *PLCG1* and *NLRC3* hub genes from the red module. (**G, H**) expression of *CD247* and *MATK* hub genes from the black module. The box presents median, 25th to 75th percentile and the whiskers present the minimum to the maximum points in the data. Comparisons were made between death (red) with survival (blue) or between HIV-negative and HIV-positive TBM by Wilcoxon rank sum test with p-values displayed as significance level above the boxes and the horizontal bars (*<0.05, **<0.01, ***<0.001).

downregulation, relative to healthy controls, were similar in PTB and TBM, although tended to be more pronounced in those with TBM as well as those with HIV (*Figure 6*).

## Transcriptional immune pathways associated with TBM mortality

To better understand the biological functions of the five modules associated with TBM mortality, a gene set associated with mortality in each module was functionally annotated using known pathway databases, such as gene ontology (GO) and Kyoto Encyclopedia of Genes and Genomes (KEGG) (*Ge et al., 2020*). Pathway enrichment analysis was performed using fold enrichment to determine if the prevalence of genes in a pathway were different from that expected by chance. We also used ssGSEA enrichment scores based on gene expression ranking of genes in a particular pathway within a single individual, to show the activity of this pathway between death and survival TBM as well as across four cohorts. These analyses helped to identify the pathways linked to mortality within the gene modules.

In the blue module, we found TBM mortality was associated with upregulated inflammatory and innate immune response transcripts, particularly in pathways involved in the acute inflammatory response and the regulation of inflammatory responses, including responses to bacteria and neutrophil activation. KEGG pathway analysis suggested this signal was associated with transduction and immune system pathways, such as toll-like receptor signaling, TNF signaling, NF-kappa B signaling and neutrophil extracellular trap formation (*Figure 5A*; *Supplementary file 1F*).

We did not find any pathway significantly associated with mortality in the upregulated cyan gene module, although in the hierarchical clustering the cyan module was highly correlated with the blue - inflammatory response module (*Figure 4A*). In contrast, analyses of the down-regulated brown, red and black modules highlighted pathways involved in adaptive immunity, predominantly those mediated by lymphocytes. These included downregulations of lymphocyte proliferation, T cell activation,

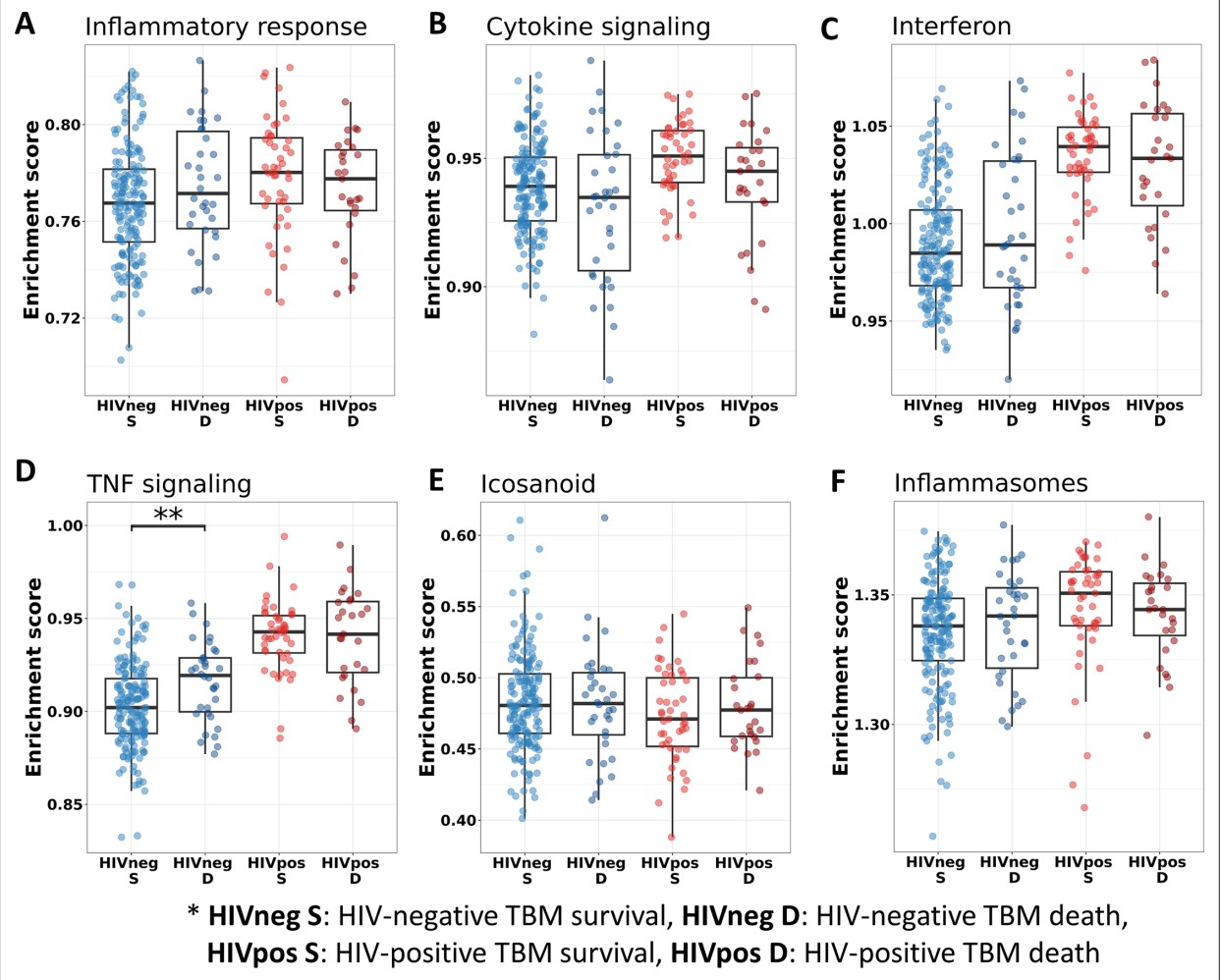

**Figure 7.** Relationship between known pathways associated with TBM pathogenesis and mortality. (**A–F**) Enrichment scores of known immune pathways associated with TBM pathogenesis. Pathway enrichment scores were calculated using single sample GSEA algorithm (**Barbie et al., 2009**). Each dot represents one participant. The box presents median, 25th to 75th percentile and the whiskers present the minimum to the maximum points in the data. The comparisons were made between survival and death using Wilcoxon rank sum test. Only significant results are presented with *<0.05, **<0.01, ***<0.001.

B cell activation, natural killer cell mediated cytotoxicity and their signaling pathways such as antigen receptor-mediated, T and B cell receptor (**Figure 5B–D**; **Supplementary file 1F**).

We also investigated the association between pathways known to be important to TB pathogenesis and mortality (**Figure 7**). Looking at those who died vs. survived amongst HIV-positive TBM, there was little difference between the groups with respect to inflammatory response, cytokine signaling, and icosanoid and inflammasome activation. This was also the case for HIV-negative individuals. However, interferon and TNF pathways were more active in those with HIV, and TNF signaling expression was significantly higher in HIV-negative adults who died rather than survived.

## HIV influence on modules, hub genes, and pathways associated with TBM mortality

Transcriptional profiles associated with TBM mortality stratified by HIV status are shown in **Figure 3**. In HIV-negative individuals, 786 (3.9%) genes were differentially expressed, whereas in HIV-positive the number was 1620 (8.1%) genes (**Figure 3B–C**; **Supplementary file 1G and H**). We hypothesized that host transcriptional signatures associated with TBM mortality differ according to HIV status. To test this hypothesis, we constructed gene co-expression networks from all genes in HIV-negative (n=207) and HIV-positive (n=74) individuals separately, then performed consensus gene co-expression

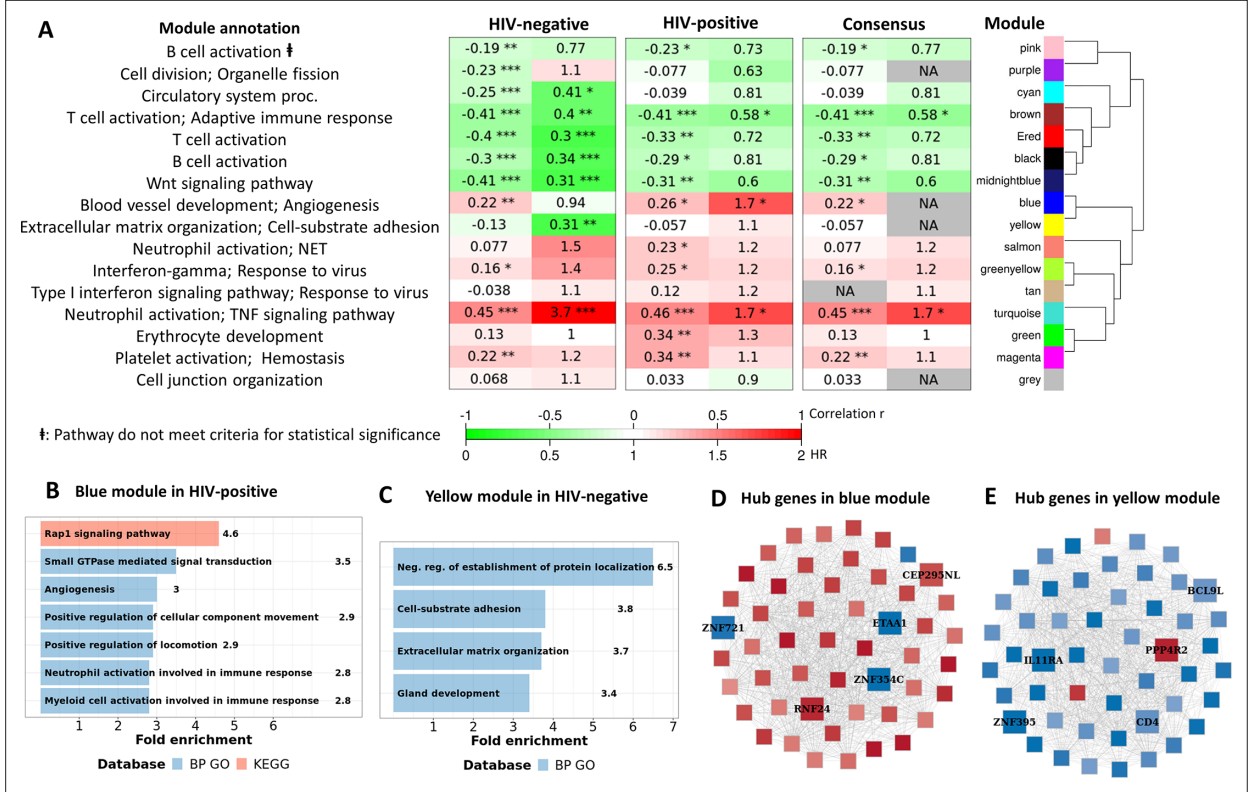

**Figure 8.** Consensus transcriptional modules associated with TBM mortality stratified by HIV-infection. (**A**) Associations between 16 consensus WGCNA modules with two clinical phenotypes TBM severity (MRC grade) and mortality in HIV-negative (n=207) and HIV-positive (n=74) TBM participants, and their associated BP Gene ontology or KEEG database. The heatmap showed the association between modules and the phenotypes, with Spearman correlation r for MRC grade and hazard ratio per increase 1/10 unit of PC1 of module (HR) for mortality in HIV-positive and HIV-negative cohorts. The consensus sub-panel presented associations of the consensus modules and clinical phenotypes with same trend detected in both HIV cohorts, otherwise were annotated with missing (NA) values. False discovery rate (FDR) corrected using Benjamini & Yekutieli procedure, with significant level denoted as *<0.05, **<0.01 and ***<0.001. Gradient colors were used to fill the cell with green indicating negative r or HR <1, red color indicating positive r or HR >1. The order of modules was based on hierarchical clustering using Pearson correlation distance for module eigengene. It is noted that these consensus modules were not identical to the identified modules in the primary analysis in *Figure 4A*. (**B–C**) Functional enrichment analysis of HIV-positive pathway (blue module) and HIV-negative pathway (yellow module), respectively. (**D–E**) Gene co-expression network of blue and yellow modules. Each node represents one gene. Each edge represents the link between two genes. Hub genes were shown by bigger nodes with bold text. The gradient color of node corresponds to its HR per 1 log$_2$ unit increase in gene expression related to mortality, with red indicating HR >1, and blue HR <1.

The online version of this article includes the following figure supplement(s) for figure 8:

**Figure supplement 1.** Construction of consensus WGCNA in HIV-negative (n=207) and HIV-positive (n=74).

**Figure supplement 2.** Gene expression of representative hub genes in healthy controls (n=30), PTB (n=295), HIV-negative TBM (n=207), and HIV-positive TBM (n=74).

network analysis (*Figure 8A*; *Figure 8—figure supplement 1*). Modules were identified with colors and to discriminate them from the modules linked to mortality alone we added an annotated function on the module name. We focused on modules that failed to form consensus associations with mortality due to opposite associations in the two cohorts.

Sixteen gene co-expression modules (*Figure 8—figure supplement 1*; *Supplementary file 1I*), ranging from 60 to 958 genes, were obtained from the HIV-negative and positive cohorts. Of these, 12 modules formed consensus association with mortality (*Figure 8A*). Of the 4 modules which failed to form consensus association, two modules were significantly associated with mortality (FDR <0.05). The blue-angiogenesis module was up-regulated in death in HIV-positive adults (HR: 1.7) and the yellow-extracellular matrix organization (EMO) module was down-regulated in death in HIV-negative adults (HR: 0.31; *Figure 8A*).

Hub genes associated with mortality, which are highly correlated with other genes in the module, were identified in the two modules. Five hub genes, with three downregulated (*ZNF354C, ZNF721,*

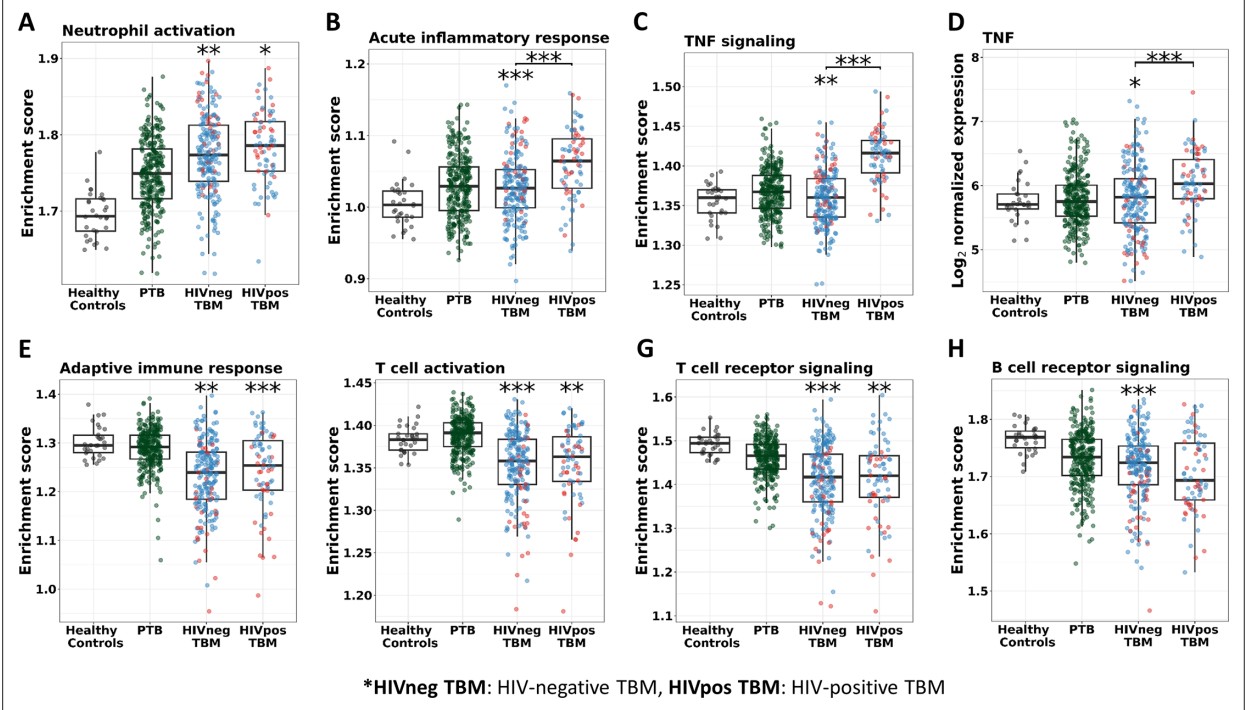

*HIVneg TBM: HIV-negative TBM, **HIVpos TBM**: HIV-positive TBM

**Figure 9.** Enrichment score of immunity pathways in healthy controls (n=30), PTB (n=295), HIV-negative TBM (n=207), and HIV-positive TBM (n=74). Pathway enrichment scores were calculated using single sample GSEA algorithm (*Barbie et al., 2009*). Each dot represents one participant. (**A–C**) showed box-plots depicting enrichment scores of the innate immunity pathways from the blue module. (**E–H**) enrichment scores of the adaptive immunity pathways from the red and brown modules and (**D**) normalized expression of TNF. The box presents median, 25th to 75th percentile and the whiskers present the minimum to the maximum points in the data. Comparisons were made between death (red) with survival (blue) or between HIV-negative and HIV-positive TBM by Wilcoxon rank sum test with p-values displayed as significance level above the boxes and the horizontal bars, respectively (*<0.05, **<0.01, ***<0.001).

The online version of this article includes the following figure supplement(s) for figure 9:

**Figure supplement 1.** Pathway fold change enrichment to mortality of top hit pathways in blue, brown and red modules from the primary analysis.

**Figure supplement 2.** Gene expression and mortality of selected hub genes in TBM, PTB and healthy participants.

**Figure supplement 3.** Performance of optimal gene set for TBM mortality prediction.

and *ETAA1*) and two upregulated (*CEP295NL* and *RNF24*), were identified in the blue-angiogenesis module. Other five hub genes, with four downregulated (*IL11RA, CD4, ZNF395,* and *BCL9L*) and one upregulated (*PPP4R2*), were identified in the yellow-EMO module (*Figure 8D–E*; *Supplementary file 1J*). Expression of some hub genes across the four cohorts showed the patterns of up (*RNF24*) and downregulation (*ZNF721, BCL9L,* and *IL11RA*), and relative to healthy controls they were similar in PTB and TBM. Expression of *RNF24* and *ZNF721* genes were significantly associated with death in HIV-positive adults, whereas expression of *BCL9L* and *IL11RA* genes were significantly associated with death in HIV-negative adults (*Figure 8—figure supplement 2*).

Pathway enrichment analysis showed genes in the blue-angiogenesis module were significantly enriched for angiogenesis or blood vessel development, leukocyte and neutrophil activation, and signal transduction pathways (*Figure 8B*, *Supplementary file 1K*). Gene expression in the yellow-module was strongly enriched in extracellular matrix organization, cell-substrate adhesion and protein localization pathways (*Figure 8C*, *Supplementary file 1*). Hierarchical clustering of modules indicated that these two modules were highly correlated (*Figure 8A*) suggesting they share similar functions, which appeared in some pathways such as angiogenesis and extracellular matrix organization.

In addition, almost all of the pathways significantly associated with TBM mortality were similar in HIV-negative and positive cohorts, confirming that these important pathways were common to all TBM (*Figure 9—figure supplement 1*). However, pathway enrichment scores were higher in HIV-positive than in HIV-negative, especially for the TNF signaling pathway and TNF transcripts (*Figure 9*). Death in HIV-negative TBM was associated with increased enrichment of these pathways, but with less

**Table 4.** Comparison of gene signatures in distinguishing survival and death in TBM prognostic models.

**RNA-seq cohorts**

| No. | Predictor set | All TBM n=281 | | HIV-negative TBM n=207 | | HIV-positive TBM n=74 | |
|---|---|---|---|---|---|---|---|
| | | AUC | Brier score | AUC | Brier score | AUC | Brier score |
| 1 | Reference model (*Thao et al., 2018*) | 0.78 | 0.14 | 0.77 | 0.12 | 0.82 | 0.18 |
| 2 | **Gene set 1** (MCEMP1, TRABD2A, and CD4) | 0.78 | 0.14 | 0.78 | 0.11 | 0.65 | 0.23 |
| 3 | **Gene set 2** (MCEMP1, NELL2 and ZNF354C) | 0.78 | 0.14 | 0.77 | 0.11 | 0.75 | 0.20 |
| 4 | **Gene set 3** (MCEMP1, TRABD2A, CD4 and ZNF354C) | 0.78 | 0.14 | 0.77 | 0.11 | 0.73 | 0.21 |
| 5 | **Gene set 4** (MCEMP1, NELL2, CD4 and ZNF354C) | 0.79 | 0.14 | 0.77 | 0.11 | 0.75 | 0.20 |
| 6 | **Gene set 3** and clinical risk factors | 0.82 | 0.13 | 0.80 | 0.11 | 0.84 | 0.15 |
| 7 | **Gene set 4** and clinical risk factors | 0.82 | 0.13 | 0.80 | 0.11 | 0.86 | 0.14 |

**qPCR validation cohort**

| No. | Predictor set | HIV-negative TBM (n=132) | |
|---|---|---|---|
| | | AUC | Brier score |
| 8* | MCEMP1, NELL2, ZNF354C and clinical predictors | 0.91 | 0.12 |

Clinical risk factors were age, MRC grade and CSF lymphocytes (*Streptomycin in Tuberculosis Trials Commitee, Medical Research Council, 1948*; *Thao et al., 2018*). The prediction models for three-month mortality were based on multivariable logistic regression models with top-hit genes and clinical risk factors. Area under the curve (AUC) and Brier score were corrected for optimism using internal bootstrap resampling over 1000 iterations to evaluate the model performance. The Brier score is an overall performance measure, calculated as the mean squared difference between the predicted probability and the actual outcome, with smaller values indicating superior model performance.

*CD4 data is unavailable for the analysis.

TNF transcript compared to HIV-positive individuals. The enrichment scores indicated mortality was strongly associated with downregulation of adaptive immune responses, including T cell activation, T and B cell receptor signaling pathways, with little impact of HIV status on these pathways (*Figure 9*). These data suggest an inadequate adaptive immune response contributes to disease pathogenesis and mortality in all those with TBM, regardless of HIV status.

## Predictors of TBM mortality

We aimed to identify baseline gene signatures that might help predict death or survival from TBM. We selected potential gene predictors from 26 common and 10 HIV-specific hub genes; these hub genes represented the dominant modules associated with mortality. Three predefined clinical factors known to be associated with outcome (age, MRC grade and CSF lymphocyte count) were also used for candidate predictor selection. Using a multivariate Cox elastic-net regression model for 39 predictors, six predictors were selected for HIV-negative TBM (MRC grade, age, CSF lymphocyte count, gene set 1: *MCEMP1*, *TRABD2A*, and *CD4*), and six predictors were selected for HIV-positive TBM (MRC grade, age, CSF lymphocyte count, gene set 2: *NELL2*, *MCEMP1*, and *ZNF354C*; *Supplementary file 1L*).

Gene expression and association with mortality of five distinct genes in gene sets 1 and 2 were presented in *Figure 9—figure supplement 2*. By combining the two gene sets above and reducing genes in the same module, we generated gene set 3 (*MCEMP1*, *TRABD2A*, *CD4,* and *ZNF354C*) and gene set 4 (*MCEMP1*, *NELL2*, *ZNF354C,* and *CD4*). These two gene sets, with and without clinical factors (MRC grade and age), were tested for their predictive performance in HIV-negative and HIV-positive RNA-seq cohorts together with a reference model (*Thao et al., 2018*). Model 7 using gene set 4 and clinical factors outperformed other gene sets and reference model, with the best overall model performance (lowest optimism-corrected Brier score, 0.11 and 0.14 for HIV-negative and HIV-positive, respectively) and the best discriminant performance (highest optimism-corrected AUC 0.80 and 0.86 for HIV-negative and positive; *Table 4*, *Figure 9—figure supplement 3*).

Given that gene expression could reflect cellular composition changes of peripheral blood with neutrophil being most abundant sub-population, we performed a sensitivity analysis including blood neutrophil count as a potential gene predictor in multivariate Cox elastic-net regression model. In sensitivity analysis, all six predictors for either HIV-negative TBM or HIV-positive TBM were repeatedly selected (*Supplementary file 1M*). We evaluated the predictive values of blood neutrophil alone and

in combination with identified gene sets, but adding neutrophils did not improve the overall performance of predictive models.

Next, we evaluated the prognostic signatures of model 7 in another sample set of 132 HIV-negative TBM patients with gene expressions measured by qPCR. CD4 was excluded due to its unavailability in model 8. The analysis demonstrated good predictive performance using model 8 with an optimism-corrected AUC 0.91 and optimism-corrected Brier score 0.12 (*Table 4*, *Figure 9—figure supplement 3*), which validating the utility of blood gene signature in early prediction of TBM mortality by qPCR assay.

## Discussion

The biological pathways involved in pathogenesis of TBM are unclear. In general, previous studies investigating TBM pathogenesis have been small, testing for relatively small numbers of selected genes or molecules, and have been unable to take an unbiased and broader view of the inflammatory response. This study investigated the pathways associated with death from TBM at a whole-genome transcriptome level in whole blood, characterizing a global dysregulation in immune responses, including inflammation, and revealing specific functional pathways and hub genes involved in TBM and the mechanisms leading to death.

We sought to understand better the pathogenesis of TBM by identifying pretreatment blood transcriptional gene modules associated TBM disease severity and three-month mortality. Four out of five identified modules were involved in immunological functions, indicating multiple functional pathways of systemic immunity are involved in the pathogenesis of TBM. In particular, mortality was strongly associated with increased acute inflammation and neutrophil activation, and decreased adaptive immunity and T and B cell activation. Whilst there appeared to be many common pathways involved in TBM mortality in HIV-positive and negative individuals, there were differences: death was associated with increased expression of angiogenesis genes in HIV-positive adults, and with TNF signaling and down regulated extracellular matrix organization in HIV-negative adults. We also identified a four-gene signature as a potential host response biomarker for mortality, regardless of HIV status.

TBM mortality was associated with increased acute inflammatory responses, the regulation of inflammatory responses, and neutrophil activation. Previous blood transcriptome studies have shown that IFN-inducible neutrophil-driven transcripts were over-expressed in blood neutrophils from active TB patients compared to those with latent TB (*Berry et al., 2010*; *Singhania et al., 2018*). Another study described that TBM adults who developed IRIS during treatment, also had significantly more abundant neutrophil-associated transcripts that preceded the development of TBM-IRIS (*Marais et al., 2017*). Our own earlier studies have also suggest a role for neutrophils in TBM pathogenesis, with high pretreatment CSF bacterial loads being correlated with high neutrophil numbers in both CSF and blood and more frequent new neurological events or paradoxical inflammatory complications (*Thuong et al., 2019*).

Taken together with previous studies, our findings support an important role for over-activation of neutrophil-mediated inflammatory responses in TBM pathogenesis and its lethal complications. Looking further into specific pathways, TBM mortality was neither associated with inflammatory response and cytokine signaling pathways in general, nor with interferon or inflammasome activation. However, increased transcripts in some specific immunity pathways, including TNF signaling, Toll-like receptor, NF-kappa B and neutrophil extracellular trap formation, were associated with TBM mortality. These pathways are involved as activators or effectors in the process of neutrophil activation and regulation, leading to an exacerbated inflammatory response (*Geng et al., 2022*; *Castro-Alcaraz et al., 2002*; *Sabroe et al., 2005*).

We found blood transcriptional responses of T cells and B cells were under-expressed in those who died from TBM, indicating an impairment in adaptive immunity in fatal disease. A reduction of activities in both T cell and B cell receptor signaling pathways in death were identified, independent of over-expression of neutrophil-mediated immune responses, indicating that multiple functional pathways influence TBM mortality. TBM pathogenesis is known to be associated with T cell impairment. Previous studies have shown lower numbers of T cells, reduced ability to respond to *Mycobacterium tuberculosis* antigens or reduced expression of activation markers and cytokine production in TBM compared to PTB and healthy individuals (*van Laarhoven et al., 2019*; *Davoudi et al., 2008*; *Shridhar et al., 2022*. This impaired T cell function has correlated with disease severity and poor

clinical outcomes in participants with PTB and TBM *van Laarhoven et al., 2019*; *An et al., 2022*; *Hemingway et al., 2017*).

B cells and antibodies also influence humoral immunity against *Mycobacterium tuberculosis*. Studies have shown a decreased memory B cell proportion, and lower levels of IgG, IgM antibodies and Fcγ receptors binding capacity in plasma in those with active lung TB compared to those with latent TB or healthy volunteers (*Lu et al., 2016*; *La Manna et al., 2022*). But little is known about the role of B cells in TBM. The observed association between TBM mortality and decreased transcriptional responses in B cell activation and B cell receptor signaling pathways suggest an unanticipated role for B cells and humoral immunity in TBM pathogenesis that needs further investigation.

Transcriptomic profiles from four cohorts, including healthy controls and PTB, provide a broad view of host responses in different TB clinical forms. Our data showed common transcriptional pathways and genes between PTB and TBM. A range of immune responses, involving in inflammation, cyto-kines, interferon, inflammasome and neutrophil signaling pathways, were activated in both PTB and TBM, but with significantly greater activation in HIV-positive TBM than HIV-negative TBM. This finding aligns with our previous data showing a dysregulated hyper-inflammation in HIV-associated TBM, with significantly higher CSF cytokine concentrations than in those without HIV infection (*Thuong et al., 2017*). These data suggest that different forms of TB are associated with similar inflammatory responses, but with different degrees of host responses, exemplified by hyper-inflammation in those with HIV.

The HIV-driven differences in TBM-associated inflammation appear sufficient to influence response to adjunctive anti-inflammatory therapy with corticosteroids. An earlier randomized controlled trial of corticosteroids in 545 predominantly HIV-negative Vietnamese adults showed they significantly improved survival (*Thwaites et al., 2004*). Our group recently completed a similar sized trial exclu-sively in HIV-positive adults without any clear benefit upon mortality (*Donovan et al., 2023*). There-fore, the differences in gene expression associated with TBM mortality in the current study – with angiogenesis activation linked to HIV-positive TBM mortality only, for example – may provide an explanation for the poor response and offer alternative therapeutic strategies. Recent studies have shown that angiogenesis is induced by *Mycobacterium tuberculosis* infection, which then contributes to inflammation, tissue damage and is correlated with disease severity (*Kumar et al., 2016*).

Developing prognostic models for TBM is important for guiding clinical decision making and improving outcomes. Several studies have developed and validated prognostic models for TBM using clinical, laboratory and radiological variables. These models have demonstrated moderate to high accuracy in predicting mortality and functional outcomes from TBM patients (*Thao et al., 2018*; *Thao et al., 2020*; *Feng et al., 2021*; *Sharawat et al., 2022*). In this study, we used blood transcrip-tional signatures and co-expression network analysis to identify module-representative hub genes. We identified a four-gene set at the start of treatment (*MCEMP1, NELL2, ZNF354C,* and *CD4*) whose expression strongly predicted three-month TBM mortality. Our prognostic models combining this four-gene host response in blood and two routine clinical predictors achieved very good performance, with AUC 0.80 and 0.86 for HIV-negative and HIV-positive individuals with TBM, respectively. This is proof-of-concept that whole blood RNA host response might be a useful pre-treatment biomarker to predict early TBM mortality. Although further investigation and validation is needed, we have identi-fied potential gene candidates for future development as prognostic biomarkers.

In summary, we present a comprehensive and unbiased analysis of the gene transcripts associated with TBM severity and mortality. Our data open a new window on TBM pathogenesis, with dysreg-ulation in both innate and adaptive immune responses strongly associated with death from TBM. Furthermore, we have identified similarities and differences in the inflammatory response associated with TBM in HIV-positive and negative adults, which may explain the different therapeutic effects of adjunctive corticosteroid treatment. We also revealed a four-gene host response signature in blood that might represent a novel biomarker for defining those at highest risk of death, regardless of their HIV status.

## Materials and methods

### Participants

We collected data and whole peripheral blood samples for transcriptomic profiling from adults (≥18 years) with TBM enrolled into two randomized controlled trials conducted in Vietnam. The two trials investigated whether adjunctive dexamethasone improves outcome from TBM and the protocols for both trials have been published (*Donovan et al., 2018b*; *Donovan et al., 2018a*). The ACT-HIV trial (NCT03092817) completed enrolment and follow-up of 520 HIV-positive adults with TBM in April 2023, with the results accepted for publication (*Donovan et al., 2023*). The LAST-ACT trial (NCT03100786) completed enrollment of 720 HIV-negative adults with TBM in March 2023, with the last participants due to complete follow-up in March 2024.

In TBM RNA-seq cohorts, peripheral blood samples were taken from 281 trial participants: the first 207 consecutively enrolled HIV-negative participants from the LAST-ACT trial, and 74 randomly selected HIV-positive participants from the ACT-HIV trial. In qPCR TBM cohorts, 132 HIV-negative participants was randomly selected from remaining participants of LAST-ACT trial with an enrichment for non-survival cases. The samples were taken at enrollment, when patients could not have received more than 6 consecutive days of two or more drugs active against M. tuberculosis. All trial participants then received standard 4-drug anti-tuberculosis treatment for 2 months, followed by 3 drugs for 10 months, and were randomly allocated to dexamethasone or identical placebo for the first 6–8 weeks (*Donovan et al., 2018b*; *Donovan et al., 2018a*). The investigators remain blind to the treatment allocation until the last participant completes follow-up and the database has been locked. Therefore, an analysis of the direct influence of corticosteroids on inflammatory response and outcome is not included in the current study. However, the metadata were approved to be unblinded but with double coded patient ID. We also blinded the treatment effect in the differential gene expression analysis. This was done by extracting only the log fold change difference between survival groups in the linear regression model, in which gene expression was outcome and survival groups and treatment were covariates.

Peripheral blood samples were taken for transcriptional profiling from two other cohorts: 295 adults with PTB and 30 healthy controls. The 295 PTB was enrolled in Pham Ngoc Thach hospital and District TB Units in Ho Chi Minh city, randomly selected from HIV-negative participants from a prospective observational study of the host and bacterial determinants of outcome from PTB (n=900) due to the very low prevalence of HIV in the study population. Participants had culture-confirmed PTB, either drug susceptible TB or a new diagnosis of multidrug-resistant TB. All participants in this cohort had <7 days of anti-TB drugs at enrolment and did not have clinical evidence of extra-pulmonary TB. Healthy controls were enrolled in Hospital for Tropical Diseases in Ho Chi Minh city, from a prospective study for epidemiological characteristics of human resistance to *Mycobacterium tuberculosis* infection. Participants in this cohort were adults without signs or symptoms of TB nor history of TB contact within the last two years.

Ethics approval was obtained from the institutional review board at the Hospital for Tropical Diseases, Pham Ngoc Thach Hospital and the ethics committee of the Ministry of Health in Vietnam, and the Oxford Tropical Research Ethics Committee, UK (OxTREC 52–16, 36–16, 24–17, 33–17 and 532–22). All participants provided their written informed consent to take part in the study, or from their relatives if they were incapacitated.

### Study design

The objectives and cohorts used in this study are presented in *Figure 1* and a workflow of data analysis is shown in *Figure 1—figure supplement 1*. Briefly, blood transcriptional profiling was generated from four cohorts of 207 TBM HIV-negative, 74 TBM HIV-positive, 279 PTB and 30 healthy controls. To define transcriptional signatures associated with TBM mortality, data from all 281 TBM participants were used. To ensure reproducibility, in our main analysis we randomly split our TBM data into two datasets, a discovery cohort (n=142) and a validation cohort (n=139). After identifying gene modules associated with mortality, related functional pathways and hub genes were determined. In a broader view, gene enrichment and expression from significant pathways and hub genes were illustrated across all four cohorts. Next, outcome prediction models were developed for TBM. Finally, the association of hub genes and outcome prediction then was validated in qPCR HIV-negative TBM cohort.

## Sample processing and RNA-seq

Whole blood samples, collected from participants at enrollment, were stored in PAXgene collection tubes at −80 °C. RNA extraction and RNA-seq were done in 2 batches. Batch 1 was done in 2020 including 207 HIV-negative TBM, 31 HIV-positive TBM and 296 PTB. Batch 2 was done in 2022 including 43 HIV-positive TBM and 30 Healthy control. RNA samples were isolated using the PAXgene Blood RNA kits (QIAGEN, Valencia, CA, USA) following the manufacturer's instructions, except for an additional washing step before RNA elution. DNA was digested on columns using the RNase-free DNase Set (QIAGEN, Valencia, CA, USA). Quality control of the RNA extraction was performed using the Epoch spec for quantity and quality, and Tapestation Eukaryotic RNA Screentape for integrity. Samples with RNA integrity number below 4 were exclude for further steps. RNA-seq was performed by the Ramaciotti Centre for Genomics (Sydney, Australia). One microgram of total RNA was used as input for each sample, using the TruSeq Stranded Total RNA Ribo-zero Globin kit (Illumina). Libraries were generated on the Sciclone G3 NGS (Perkin Elmer, Utah, USA) and the cDNA was amplified using 11 PCR cycles. Libraries were pooled 75 samples per pool and sequenced using NovaSeq 6000 S4 reagents at 2x100 bp to generate about 30 million reads per sample.

## RNA-seq data quality control and pre-processing

Quality control and alignment were performed using an in-house pipeline modified from previously published practices for RNA-seq analysis (*Yalamanchili et al., 2017*; *Conesa et al., 2016*) in linux command line. Briefly, the quality of the sequencing fastq files was analyzed using FastQC (v0.11.5) and poor quality samples were excluded from further analysis. Sequence reads were adapter and quality trimmed using Trimmomatic (v0.36), followed by duplicated optical read removal using BBMap (v38.79) tool. STAR aligner (v2.5.2a) was used to align the reads to the human reference genome (GRCh38 build 99) downloaded from Ensembl, allowing for maximum 2 mismatches in each 25 bp segment and a maximum of 20 alignment hits per read (*Dobin et al., 2013*; *Bolger et al., 2014*). The alignment results were sorted and indexed for downstream analyses as BAM format files. The aligned reads were further utilized to generate gene expression counts using FeatureCounts (v2.0.0) against the human reference annotation (GRCh38 build 99; *Liao et al., 2014*). Next, 60,067 genes in the expression matrix were first normalized by variance stabilizing transformation method built in DESeq2 package in R (*Love et al., 2014*). Subsequently, the batch effect was removed for 20,000 most variable genes using combat function in the SVA package which corrected for 2 RNA-seq batches (*Leek et al., 2012*). The results of batch effect removal were visualized by principle component analysis. First component was plotted against second component. The variation explained by RNA-seq batches was removed after using combat (*Figure 1—figure supplement 2*). All later analysis and data visualization were done used batch corrected data.

## WGCNA preservation analysis on discovery and validation dataset

In this study, WGCNA from Bioconductor R package (version 4.3.3) was used to construct weighted genes co-expression network in discovery dataset (n=139). In brief, 20,000 most variant genes out of 60,067 genes in the expression matrix were first selected and normalized by variance stabilizing transformation method built in DESeq2 package in R. The batch effect was removed on normalized expression data using combat function in the SVA package. After removing batch effect, 5000 most variable genes across 281 TBM were input in unsupervised principle component analysis to check for potentially outlying individuals. Specifically, we declared individuals as outliers if they were >2.5 standard deviations away from the mean first or second principle component. No individual was classified as outliers using these criteria. Using WGCNA, the similarity matrix between each pair of genes across all samples was calculated based on its Pearson's correlation value. Then, the similarity matrix was transformed into an adjacency matrix. Subsequently, the topological overlap matrix (TOM) and the corresponding dissimilarity (1-TOM) value were computed based on soft threshold $\beta = 8$ which meet the criteria for scale-free topology property of the co-expression network ($R^2$-cutoff for scale-free topology $= 0.85$). Finally, a dynamic tree cut algorithm with deepSplit of 2, cut height of 0.975 and minimum module size of 30 genes, was employed to detect gene co-expression modules, groups of genes with a similar expression. The expression patterns of each gene module were summarized by the first principle component (PC1). Pairs of modules were subsequently merged if the correlation

between the modules' PC1 exceeded 0.7. Genes that did not fit the clustering criteria were combined in a leftover group named grey.

If a module in the discovery dataset is not determined randomly, it will be reproduced in validation dataset. In this study, module preservation statistics was used to validate whether a defined module in discovery dataset could also be found in validation dataset. The WGCNA used two composite preservation statistics for module preservation: First, Z-summary distinguished preserved modules from non-preserved ones through the permutation test (n Permutations = 1000). Values below 2 of Z-summary indicate non-preserved modules, while values over 2 represent moderately preserved modules, and values over 10 provides strong evidence of module preservation. Next, median ranks were computed. In comparisons of the two modules, the one with a higher median rank was considered to have a lower preservation tendency (*Langfelder et al., 2011*).

## WGCNA consensus analysis in HIV-negative and HIV-positive TBM cohorts

Similar to preservation analysis, 5000 most variance genes were input in WGCNA consensus analysis. The similarity matrix between each pair of genes across all samples was calculated based on its Pearson's correlation value. Then, the similarity matrix was transformed into an adjacency matrix. Subsequently, the topological overlap matrix (TOM) and the corresponding dissimilarity (1-TOM) value were computed based on soft threshold $\beta = 8$ which meet the criteria for scale-free topology property of the co-expression network. Considering the diverse statistical properties between different HIV-negative and HIV-positive data sets, we scaled and transformed the HIV-positive TOM to make it equivalent to that HIV-negative and obtained consensus modules between two TBM cohorts; the consensus TOM was calculated with component-wise ('parallel') minimum of the TOMs for each set. To obtain large modules, the 'minModuleSize' parameter, indicating the minimum module size of the modules, was set as 30. Genes with similar expression patterns were separated into different modules with the 'cutreeDynamic' function; to evaluate and group the co-expression similarities of all modules, the eigengenes (MEs) were calculated, clustered, and mapped to the related consensus modules; then, modules with a correlation of 0.75 were merged with 'mergeCloseModules' function using default parameters.

## Validation of hub genes by Microfluidic multiplex RT-qPCR

Whole blood samples were collected, stored and RNA extraction was performed as described in previous section. The expression of housekeeping genes (GAPDH and TMBIM6) and other hub genes were evaluated by microfluidic RT-qPCR using Biomark 48.48 Complete Bundle with Delta Gene Assays and BioMark HD system (Fluidigm Corporation, South San Francisco, CA, USA) following manufacturer's instructions with some optimized modifications. Briefly, 2 µL of total RNA with concentration of 50 ng/µL was reverse transcribed to cDNA. The specific target amplification (STA) of cDNA was used for 14 cycles of preamplification, then the STA products were treated with Exonuclease I (New England Biolabs, Ipswich, MA, USA) before 20-fold dilution. SsoFast EvaGreen Supermix with Low ROX (Bio-Rad Laboratories, Hercules, CA, USA) was used in RT-qPCR before applying to IFC Controllers MX and BioMark HD system (Fluidigm Corporation, South San Francisco, CA, USA). The CT values of target hub genes were normalized before analysis.

## Statistical analysis

The primary outcome examined in this study was TBM three-month mortality. In the descriptive analysis, we summarized and tested association of patient characteristics with three-month mortality using univariate Cox regression analysis. We presented the proportion for binary variables and median (1st and 3rd interquartile range) for continuous variables.

To explore transcriptional profiles associated with mortality in TBM, differential expression analysis was performed on the 20,000 normalized gene expression matrix to find differentially expressed genes (DEGs). We constructed the contradicted matrix with survival and death status adjusted for covariates including: age, HIV status, corticosteroid treatment (LAST-ACT trial investigators remained blind). Linear models were used to assess DEGs using limma R package. Empirical Bayes moderated-t p-values were computed for each genes and Benjamini-Hochberg were used for correcting multiple testing (FDR). We defined the DEGs with the parameters (fold change >1.5 and FDR <0.05).

To visualize samples clustering in the four cohorts, an unsupervised principal component analysis was performed on 20,000 normalized genes, and the first principle component was plotted against the second principle component. The 95% confidence ellipse was drawn for each group using multivariate t-distribution ellipse. To compare enrichment activity of interested pathways between death vs survival, TBM vs PTB or healthy controls, enrichment scores of these pathways were calculated for single patient using single sample Gene Set Enrichment Analysis (GSEA) algorithm (ssGSEA; *Barbie et al., 2009*). Pairwise Wilcoxon rank sum test was used for comparison between any two cohorts.

In the primary analysis, we initially conducted *weighted gene co-expression network analysis* (WGCNA) on the whole-blood transcriptomic profiling of the discovery cohort to identify clusters of genes (or 'modules') in the gene co-expression network among host transcriptomic genes. We then performed a *network modules preservation analysis* to assess the reproducibility of these modules and the network in the validation cohort. Subsequently, we conducted a *module-trait association analysis* to identify modules associated with the clinical traits of mortality and TBM severity. For the association analysis with mortality, we used a *Cox regression model* with the PC1 of the module as an independent variable. We adjusted the model for age, HIV status, and corticosteroid treatment.

To analyze the association with TBM severity, we calculated the *Spearman correlation* between TBM severity (MRC grade) and the module's PC1. We corrected for false discovery rate (FDR) for the dependent hypotheses by using the Benjamini-Yekutieli procedure. Target modules were defined as modules associated with mortality in both the discovery and validation cohorts at FDR <0.05. Hub genes within each targeted module were identified based on three criteria: being protein coding gene, module membership cut off 0.85 (MM) and high rank of gene significance rank (GS). MM defined as correlation between individual gene and the module's PC1. GS defined as -$\log_{10}$ p value association of gene with mortality using Cox regression model adjusted for age, HIV status and dexamethasone treatment. Genes were first filtered for protein coding function and MM above 0.85. Genes then was ranked based on its GS and top 20 hub genes were selected if number of genes in module above 500, otherwise top 10 genes were selected. Overlap hub genes between discovery and validation cohort were consider validated.

In the subsequent analysis, to determine the biological functions or processes potentially related to each module we conducted *overrepresentation analysis* (ORA) using ShinyGO v0.77 (*Ge et al., 2020*). Genes were first filters for association with three-month mortality at p<0.05 and then were input into ShinyGO. We used both *GO and KEGG databases* for this analysis. We pre-specified significant pathways with overlap gene in pathway >5 and FDR <0.05.

Furthermore, to compare *enrichment activity* of resulting pathways between death vs survival, TBM vs PTB or healthy controls, enrichment scores of these pathways were calculated for single patient using ssGSEA algorithm (*Barbie et al., 2009*). Pairwise Wilcoxon rank rum test were applied for comparison between two cohorts. To compare or visualize the difference of pathway effect on mortality between HIV-negative and HIV-positive TBM, *fold change enrichment* of interested pathways were calculated using Quantitative Set Analysis for Gene Expression (QuSAGE) method (*Yaari et al., 2013*).

In the secondary analysis, we performed a *consensus* WGCNA analysis to identify consensus patterns of co-expression networks between HIV-positive and HIV-negative conditions in our TBM cohort. We conducted similar association analyses for TBM severity and mortality as in the preceding WGCNA analysis for both the HIV-negative and HIV-positive cohorts. We visualized the results using a heatmap. We focused on modules that failed to form consensus associations with mortality due to opposite associations in the two cohorts and defined as HIV-positive-specific signals and HIV-negative-specific signals. Additionally, the functional enrichment ORA was performed for these consensus modules and the HIV-specific modules. Top 5 hub genes from HIV-specific modules were selected based on criteria described in primary analysis, which included being protein-coding genes and having high MM and high rank GS. GS in this analysis was calculate as -$\log_{10}$ p value association of gene with mortality using Cox regression model adjusted for age and corticosteroid treatment.

We then performed variable selection using a *multivariate elastic-net Cox regression model* to select the important predictors for HIV-negative and HIV-positive TBM prognosis separately. Candidate predictors in HIV-negative TBM consisted of the combination of common hub genes, HIV-negative specific hub genes and clinical biomarkers: age, TBM severity and CSF lymphocyte count. Similarly, for HIV-positive TBM, candidate predictors consisted of common hub genes, HIV-positive

specific hub genes and clinical biomarkers. We performed the analysis with 1,000 bootstrap sampling with replacement. The chosen variables were (1) among the top 75% of selected variables and (2) one representative hub gene per module. Subsequently, we developed a prediction model based on a *logistic regression model* incorporating only the chosen variables for all TBM cohort or stratified by HIV-negative and HIV-positive cohort. We also assessed predictive performance of chosen variables in each cohort by 1000 times bootstrapping sampling approach and reported the overall model performance *optimism-corrected Brier score*, the discrimination (*optimism-corrected AUC*), and calibration (*optimism-corrected calibration-slope*) of the developed model (*Steyerberg et al., 2001*).

## Acknowledgements

We thank and the clinical staff who recruited patients into our study from Hospital for Tropical Diseases and Pham Ngoc Thach hospital, Ho Chi Minh city, Vietnam, and all participants. This work was supported by Wellcome Trust Fellowship in Public Health and Tropical Medicine to NTTT (206724/Z/17/Z), Wellcome Trust Investigator Award to GT and Wellcome Trust to Vietnam Africa Asia Programme.

## Additional information

### Funding

| Funder | Grant reference number | Author |
|---|---|---|
| Wellcome Trust | 10.35802/206724 | Nguyen Thuy Thuong Thuong |
| Wellcome Trust | 10.35802/106680 | Guy E Thwaites |
| Wellcome Trust | 10.35802/110179 | Guy E Thwaites |

The funders had no role in study design, data collection and interpretation, or the decision to submit the work for publication. For the purpose of Open Access, the authors have applied a CC BY public copyright license to any Author Accepted Manuscript version arising from this submission.

### Author contributions

Hoang Thanh Hai, Formal analysis, Visualization, Writing – original draft, Writing – review and editing; Le Thanh Hoang Nhat, Supervision, Methodology, Writing – original draft, Writing – review and editing; Trinh Thi Bich Tram, Validation, Visualization, Methodology, Writing – review and editing; Do Dinh Vinh, Validation, Visualization, Methodology; Artika P Nath, Formal analysis, Supervision, Writing – review and editing; Joseph Donovan, Nguyen Thi Anh Thu, Dang Van Thanh, Nguyen Duc Bang, Dang Thi Minh Ha, Nguyen Hoan Phu, Ho Dang Trung Nghia, Le Hong Van, Data curation, Project administration; Michael Inouye, Conceptualization, Supervision, Methodology, Writing – review and editing; Guy E Thwaites, Conceptualization, Supervision, Funding acquisition, Methodology, Writing – review and editing; Nguyen Thuy Thuong Thuong, Conceptualization, Supervision, Funding acquisition, Methodology, Writing – original draft, Writing – review and editing

### Author ORCIDs

Hoang Thanh Hai ⓘ https://orcid.org/0000-0002-1809-5541
Michael Inouye ⓘ https://orcid.org/0000-0001-9413-6520
Guy E Thwaites ⓘ https://orcid.org/0000-0002-2858-2087
Nguyen Thuy Thuong Thuong ⓘ https://orcid.org/0000-0001-8733-692X

### Ethics

Clinical trial registration NCT03100786; NCT03092817.
Ethics approval was obtained from the institutional review board at the Hospital for Tropical Diseases, Pham Ngoc Thach Hospital and the ethics committee of the Ministry of Health in Vietnam, and the Oxford Tropical Research Ethics Committee, UK (OxTREC 52-16, 36-16, 24-17, 33-17 and 532-22). All participants provided their written informed consent to take part in the study, or from their relatives if they were incapacitated.

Reviewer #1 (Public review): https://doi.org/10.7554/eLife.92344.3.sa1
Reviewer #2 (Public review): https://doi.org/10.7554/eLife.92344.3.sa2
Author response https://doi.org/10.7554/eLife.92344.3.sa3

## Additional files

### Supplementary files
• Supplementary file 1. Additional analysis and results.
• MDAR checklist

### Data availability
All source data for figures and tables, as well as the source code, are deposited in Dryad https://doi.org/10.5061/dryad.s4mw6m9gf.

The following dataset was generated:

| Author(s) | Year | Dataset title | Dataset URL | Database and Identifier |
|---|---|---|---|---|
| Nguyen et al. | 2024 | Whole blood transcriptional profiles and the pathogenesis of tuberculous meningitis | https://doi.org/10.5061/dryad.s4mw6m9gf | Dryad Digital Repository, 10.5061/dryad.s4mw6m9gf |

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
