## [Editor Report · eLife assessment]

In this **valuable** study, the authors investigate the transcriptional landscape of tuberculous meningitis. They reveal potentially significant molecular differences contributed by HIV co-infection, and derive a prognostic model to predict mortality combining a gene expression signature with clinical parameters. Whilst some of the evidence presented is **compelling**, the bioinformatics analysis remains limited and cannot be used to make causal inferences and conclusions about immunopathogenesis for tuberculous meningitis. The work will be of broad interest to the infectious disease community however, further validation of the findings is critical for future utility.

---

## [Referee Report · Reviewer #1 (Public review)]

Summary:

Tuberculous meningitis (TBM) is one of the most severe form of extrapulmonary TB. TBM is especially prevalent in people who are immunocompromised (e.g. HIV-positive). Delays in diagnosis and treatment could lead to severe disease or mortality. In this study, the authors performed the largest ever host whole blood transcriptomics analysis on a cohort of 606 Vietnamese participants. The results indicated that TBM mortality is associated with increased neutrophil activation and decreased T and B cell activation pathways. Furthermore, increased angiogenesis was also observed in HIV-positive patients who died from TBM, whereas activated TNF signaling and down-regulated extracellular matrix organisation were seen in the HIV-negative group. Despite similarities in transcriptional profiles between PTB and TBM compared to healthy controls, inflammatory genes were more active in HIV-positive TBM. Finally, 4 hub genes (MCEMP1, NELL2, ZNF354C and CD4) were identified as strong predictors of death from TBM.

Strengths:

This is a really impressive piece of work, both in terms of the size of the cohort which took years of effort to recruit, sample and analyse and also the meticulous bioinformatics performed. The biggest advantage of obtaining a whole blood signature is that it allows an easier translational development into test that can be used in the clinical with a minimally invasive sample. Furthermore, the data from this study has also revealed important insights in the mechanisms associated with mortality and the differences in pathogenesis between HIV-positive and HIV-negative patients, which would have diagnostic and therapeutic implications.

Weaknesses:

The authors have addressed all the weaknesses in the revised version.

---

## [Referee Report · Reviewer #2 (Public review)]

Summary:

This manuscript describes the analysis of blood transcriptomic data from patients with TB meningitis, with and without HIV infection, with some comparison to those of patients with pulmonary tuberculosis and healthy volunteers. The objectives were to describe the comparative biological differences represented by the blood transcriptome in TBM associated with HIV co-infection or survival/mortality outcomes, and to identify a blood transcriptional signature to predict these outcomes. The authors report an association between mortality and increased levels of acute inflammation and neutrophil activation, but decreased levels of adaptive immunity and T/B cell activation. They propose a 4-gene prognostic signature to predict mortality.

Strengths:

Biological evaluations of blood transcriptomes in TB meningitis and their relationship to outcomes have not been extensively reported previously.

The size of the data set is a major strength and is likely to be used extensively for secondary analyses in this field of research.

The addition of a new validation cohort to evaluate the generalisability of their prognostic model in the revised manuscript is welcome.

Weaknesses:

The bioinformatic analysis is limited to a descriptive narrative of gene-level functional annotations curated in GO and KEGG databases. This analysis cannot be used to make causal inferences. In addition the functional annotations are limited to 'high-level' terms that fail to define the biology very precisely. As a result, the conclusions about the immunopathogenesis of TBM are not adequately substantiated.

The lack of AUROC confidence intervals and direct comparison to the reference prognostic model in the validation cohort undermines confidence in their conclusion that their new prognostic model combing gene expression data and clinical variables performs better than the reference model.

---

## [Author Response]

The following is the authors’ response to the original reviews.

**eLife assessment**
In this valuable study, the authors investigate the transcriptional landscape of tuberculous meningitis, revealing important molecular differences contributed by HIV co-infection. Whilst some of the evidence presented is compelling, the bioinformatics analysis is limited to a descriptive narrative of gene-level functional annotations, which are somewhat basic and fail to define aspects of biology very precisely. Whilst the work will be of broad interest to the infectious disease community, validation of the data is critical for future utility.

We appreciate with eLife’s positive assessment, although we challenge the conclusion that we ‘fail to define aspects of biology very precisely’. Our stated objective was to use bioinformatics tools to identify the biological pathways and hub genes associated with TBM pathogenesis and the eLife assessment affirms we have investigated ‘the transcriptional landscape of tuberculous meningitis’. To more precisely define aspects of the biology will require another study with different design and methods.

**Reviewer #1 (Public Review):**
Summary:Tuberculous meningitis (TBM) is one of the most severe forms of extrapulmonary TB. TBM is especially prevalent in people who are immunocompromised (e.g. HIV-positive). Delays in diagnosis and treatment could lead to severe disease or mortality. In this study, the authors performed the largest-ever host whole blood transcriptomics analysis on a cohort of 606 Vietnamese participants. The results indicated that TBM mortality is associated with increased neutrophil activation and decreased T and B cell activation pathways. Furthermore, increased angiogenesis was also observed in HIV-positive patients who died from TBM, whereas activated TNF signaling and down-regulated extracellular matrix organisation were seen in the HIV-negative group. Despite similarities in transcriptional profiles between PTB and TBM compared to healthy controls, inflammatory genes were more active in HIV-positive TBM. Finally, 4 hub genes (MCEMP1, NELL2, ZNF354C, and CD4) were identified as strong predictors of death from TBM.Strengths:This is a really impressive piece of work, both in terms of the size of the cohort which took years of effort to recruit, sample, and analyse, and also the meticulous bioinformatics performed. The biggest advantage of obtaining a whole blood signature is that it allows an easier translational development into a test that can be used in the clinical with a minimally invasive sample. Furthermore, the data from this study has also revealed important insights into the mechanisms associated with mortality and the differences in pathogenesis between HIV-positive and HIV-negative patients, which would have diagnostic and therapeutic implications.Weaknesses:The data on blood neutrophil count is really intriguing and seems to provide a very powerful yet easy-to-measure method to differentiate survival vs. death in TBM patients. It would be quite useful in this case to perform predictive analysis to see if neutrophil count alone, or in combination with gene signature, can predict (or better predict) mortality, as it would be far easier for clinical implementation than the RNA-based method. Moreover, genes associated with increased neutrophil activation and decreased T cell activation both have significantly higher enrichment scores in TBM (Figure 9) and in morality (Figure 8). While I understand the basis of selecting hub genes in the significant modules, they often do not represent these biological pathways (at least not directly associated in most cases). If genes were selected based on these biologically relevant pathways, would they have better predictive values?

We conducted a sensitivity analysis including blood neutrophil as a potential predictor in the multivariate Cox elastic-net regression model for important predictor selection (Table S14). In this analysis, all six selected important predictors (genes and clinical risk factors) identified in the original analysis (Table S13) were also selected, together with blood neutrophil number. Additionally, we evaluated the predictive value of blood neutrophil alone, which demonstrated poor performance, with an optimism-corrected AUC of 0.63 for all TBM, 0.67 for HIV-negative TBM, and 0.70 for HIV-positive TBM. Even when combined with identified gene signatures, blood neutrophil did not improve the overall performance of predictive model (optimism-corrected AUC of 0.79 for all TBM, 0.76 for HIV-negative TBM, and 0.80 for HIV-positive). These results indicate that identified hub genes exhibit better predictive values compared to blood neutrophil alone or in combination. These findings have been incorporated into our manuscript results.

To test whether pathway representative genes have better predictive values than hub genes, we included all these genes in the analysis for important predictor selection. Pathway representative genes comprised ANXA3 and CXCR2 representing neutrophil activation and IL1b representing acute inflammatory response. We observed that all hub genes (MCEMP1, NELL2, ZNF354C, and CD4) consistently emerged as the most important genes with the highest selection in the models, compared to the rest, in both the HIV-negative TBM and HIV-positive TBM cohorts. Additionally, these identified hub genes were still selected when testing together with other hub genes representing relevant biological pathways associated with TBM mortality, such as CYSTM1 involved in neutrophil activation, TRAF5 involved in NF-kappa B signaling pathway, CD28 and TESPA1 involved in T cell receptor signaling. These results show that selected genes based on known biologically relevant pathways did not give better predictive values than the identified hub genes in the significant modules.

**Reviewer #2 (Public Review):**
Summary:This manuscript describes the analysis of blood transcriptomic data from patients with TB meningitis, with and without HIV infection, with some comparison to those of patients with pulmonary tuberculosis and healthy volunteers. The objectives were to describe the comparative biological differences represented by the blood transcriptome in TBM associated with HIV co-infection or survival/mortality outcomes and to identify a blood transcriptional signature to predict these outcomes. The authors report an association between mortality and increased levels of acute inflammation and neutrophil activation, but decreased levels of adaptive immunity and T/B cell activation. They propose a 4-gene prognostic signature to predict mortality.Strengths:Biological evaluations of blood transcriptomes in TB meningitis and their relationship to outcomes have not been extensively reported previously.The size of the data set is a major strength and is likely to be used extensively for secondary analyses in this field of research.Weaknesses:The bioinformatic analysis is limited to a descriptive narrative of gene-level functional annotations curated in GO and KEGG databases. This analysis cannot be used to make causal inferences. In addition, the functional annotations are limited to 'high-level' terms that fail to define biology very precisely. At best, they require independent validation for a given context. As a result, the conclusions are not adequately substantiated. The identification of a prognostic blood transcriptomic signature uses an unusual discovery approach that leverages weighted gene network analysis that underpins the bioinformatic analyses. However, the main problem is that authors seem to use all the data for discovery and do not undertake any true external validation of their gene signature. As a result, the proposed gene signature is likely to be overfitted to these data and not generalisable. Even this does not achieve significantly better prognostic discrimination than the existing clinical scoring.

As explained in response to the eLife assessment, our objective was to use bioinformatics tools to identify the biological pathways and hub genes associated with TBM pathogenesis. We agree that ‘This analysis cannot be used to make causal inferences’: that would require different study design and approaches. The proposed gene signature has higher AUC values than the existing clinical model alone or in combination with clinical risk factors (Table 4). We agree that independent validation of the gene signature will be a crucial next step for future utility. We have performed qPCR in another sample set, and have added these results in the revision (Table 4 and supplementary figure S8)

**Reviewer #1 (Recommendations For The Authors):**
I have a few additional comments most of which are relatively minor:(1) Can the authors please clarify if all the PTB cases are also HIV-negative?

This has been added to the methods section.

(2) For Table 1, can the authors please list the total number of patients with microbiologically confirmed TB regardless of the methods used? And for the two TBM groups, was the positive microbiology based on CSF findings?

The total number of patients with microbiologically confirmed TB was presented in Table 2 in definite TBM group, which was microbiologically confirmed TB diagnosed using microscopy, culture, and Xpert testing in cerebrospinal fluid (CSF) samples. We have updated the note in Table 2 to provide clarity on the definition.

(3) How was the discovery and validation set selected? Was it based on randomisation?

We randomly split TBM data into two datasets, a discovery cohort (n=142) and a validation cohort (n=139) with a purpose to ensure reproducibility of data analysis. We described this in the methods section.

(4) Line 107 can be better clarified by stating that the overall 3-month mortality rate is 21.7% for TBM regardless of HIV status.

Thank you, we have restated this sentence in the results section.

(5) The authors stated that samples were collected at enrolment when patients would have received less than 6 days of anti-tubercular treatment. Is there information on the median and IQR on the number of days that the patients would have received Rx, especially between the groups? Did the authors control for this variable when analysing for DEGs?

One of criteria to enroll participants in LAST-ACT and ACT-HIV trials is that they must receive less than 6 consecutive days of two or more drugs active against M. tuberculosis. However, the information of the days that the patients would have received Rx was not recorded and we could not control this variable when performing differential expression analysis for DEGs. This has been clarified further in the methods section: ‘The samples were taken at enrollment, when patients could not have received more than 6 consecutive days of two or more drugs active against M. tuberculosis.’

(6) I am a little bit concerned with the reads mapping accuracy (57%) to the human genome, which is fairly low. Did the authors investigate the reasons behind this low accuracy?

Thank you. It was indeed a typo. We have corrected it in the results section.

(7) On Tables S2-S4, can the authors please clarify what the last column (labelled as "B") shows?

Tables S2-S4 now have been changed to S3-S5. We have updated the legend of these tables to provide clarification regarding the meaning of the last column.

**Reviewer #2 (Recommendations For The Authors):**
If the authors wish to revise their manuscript, I suggest the following amendments:(1) Provide a consort diagram for the selection of samples included in the present analysis (from parent study cohorts), allocation to test and validation splits for bioinformatics analysis, and outcomes.

We have provided our consort diagram in supplementary Figure S10.

(2) Provide details of inclusion criteria for pulmonary TB cohort, and how samples from this cohort were selected for inclusion in the present analysis. Please clarify whether this cohort excluded HIV-positive participants by design or by chance.

The inclusion criteria for the pulmonary TB cohort were described in the methods section. Due to the very low prevalence of HIV in this prospective observational study, HIV-positive participants were excluded. We have clarified in the amended manuscript that the pulmonary TB cohort only included HIV-negative participants.

(3) Baseline characteristics of HIV-positive participants (Table 1) should include CD4 count, HIV viral load, and whether anti-retroviral therapy was naïve or experienced.

We have included pre-treatment CD4 cell count, information on anti-retroviral therapy, and HIV viral load data in Table 1, as well as described these information in the results section.

(4) I note that the TBM samples were derived from RCTs of adjunctive steroid therapy, but not stratified in the present analysis by treatment arm allocation. Clearly, this may affect the survival/mortality outcomes that are the central focus of this manuscript. Therefore, they should be included in the models for differential gene expression analysis and prognostic signature discovery. To do so, the authors may need to wait until they are able to unblind the trial metadata.

With permission from the trial investigators, we were able to adjust the analyses for treatment with corticosteroids. The investigators remained blind to the allocation and we have not reported any direct effects of corticosteroids on outcome – such an analysis could only be done once the LAST-ACT trial has been reported (which won’t be until the end of 2024). Treatment outcome and effect were blinded by extracting only the fold change difference between survival and death in the linear regression model, in which gene expression was outcome and survival and treatment were covariates.

(5) I understood from the methods (lines 460-461) that batch correction of the RNAseq data was necessary. However, it is not clear how the samples were batched. PCA of the transcriptomes before and after batch correction with batch and study group labels should be provided. I would also advocate for a sensitivity analysis to check the robustness of the main findings without batch correction. I assume Fig2A represents batch-corrected data, but this is not clear.

We have now added information about the RNA sequencing batch and the batch correction approach, analyses and data visualizations utilized batch-corrected data in the methods section. We have also updated results related to batch correction in Fig. 2A and Supplementary Figure S9.

(6) I would encourage the authors to include a differential gene expression analysis to directly compare the transcriptome of TBM to that of pulmonary TB. I think it would add additional value to their focus on describing the transcriptome in TBM.

We thank for reviewer’s suggestion. Conducting differential gene expression analysis to compare the transcriptome of TBM with that of PTB is beyond the scope of this manuscript and we will examine this question separately.

(7) I don't really understand the purpose of splitting their data set into test and validation for the purposes of showing that WGCNA analysis is mostly reproduced in the two halves of the data. I would advocate that they scrap this approach to maximise the statistical power of their analysis in the descriptive work.

As mentioned in response to reviewer #1 in question #3, the purpose of splitting data is to ensure the reproducibility of the data analysis as suggested by Langfelder et al. (PMID: 21283776). This approach served two purposes: (i) to affirm the existence of functional modules in an independent cohort and (ii) to validate the association of interested modules or their hub genes with survival outcomes.

(8) The authors should soften the confidence in their interpretation of the GO/KEGG annotations of WGCNA modules. At least, they should include a paragraph that explicitly details the limitations of their analyses, including (i) the accuracy GO/KEGG annotations are not validated in this context (if at all), (ii) that none of the data can be used to make causal inferences and (iii) that peripheral blood assessments that are obviously impacted by changes in cellular composition of peripheral blood do not necessarily reflect immunopathogenesis at the site of disease - in fact if circulating cells are being recruited to the site of disease or other immune compartments, then quite the opposite interpretations may be true.

We appreciate the reviewer's comment. (i) In our analysis, we initially confirmed the existence of Weighted Gene Co-expression Network Analysis (WGCNA) modules in discovery cohort and validated the association of these modules with mortality outcomes in validation cohort. We then applied GO/KEGG annotations to define the biological functions involved in WGCNA modules. Finally, we performed Qusage analysis to directly test the association of top-hit pathways of each WGCNA module with mortality outcomes (see supplementary S6). This analysis approach helped to identify and validate modules and biological pathways associated with TBM mortality in this context, avoiding potential false positives in GO/KEGG annotations of WGCNA modules. (ii) We agree with the assessment that 'This analysis cannot be used to make causal inferences,' as that would require a different study design and approach. (iii) The focus of this study is to investigate the pathogenesis of TBM in the systemic immune system. We have highlighted this focus in the title and the aim of the manuscript.

(9) For the prognostic signature discovery and validation, I strongly recommend the authors include more robust validation. For example, to undertake an 80:20 split for sequential discovery (for feature selection and derivation of a prognostic model), followed by validation of a 'locked' model in data that made no contribution to discovery. In two separate sensitivity analyses. I also suggest they split their dataset (i) by treatment allocation in the RCT and (ii) by HIV status. In addition, their method for feature selection has to be clearer- precisely how they select hub genes from their WGCNA analysis as candidate predictors is not explained. Since this is such a prominent output of their manuscript, the results of this analysis should really be included in the main manuscript, and all performance metrics for discrimination should include confidence intervals.

Employing an 80:20 split for training and testing models is a good approach for an internal validation. However, we addressed the issue of overestimating the performance of a prognostic model by bootstrapping sampling approach proposed by Steyerberg et al. (PMID: 11470385). This approach has been proven to provide stable estimates with low bias. The overall model performance for discrimination, reported in our manuscript, was corrected for “optimism” to ensure internal validity. This adjustment was achieved through a 1000-times bootstrapping approach, which effectively accounted for estimation uncertainty. As such, there is no need to present confidence intervals for these metrics.

Moreover, in our revision, to confirm prognostic signatures independently, we have evaluated the predictive value of identified gene signatures using qPCR in another set of samples. The results have been added in Table 4, supplementary Figure S8 and the results section.

For the reasons given above (comment 4), we are unable to split our dataset by treatment allocation in this analysis. But as described, we have adjusted the analysis for corticosteroid treatment. Once the primary results of the LAST ACT trial have been published, we will examine the impact of corticosteroids on TBM pathophysiology and outcomes, seeking to better understand the mechanisms by which steroids have their therapeutic effects.

Given the difference in pathogenesis and immune response by HIV-coinfection, we stratified our analysis by HIV status. As the reviewer’s suggestion, we have provided additional details in the methods section regarding the selection of hub genes from associated WGCNA modules and the feature selection process for predictive modeling.